# R-loops acted on by RNase H1 influence DNA replication timing and genome stability in *Leishmania*

Jeziel D. Damasceno ®[1] ✉, Emma M. Briggs ®[2,3], Marija Krasilnikova ®[1], Catarina A. Marques ®[1], Craig Lapsley[1] & Richard McCulloch ®[1] ✉

Genomes in eukaryotes normally undergo DNA replication in a choreographed temporal order, resulting in early and late replicating chromosome compartments. *Leishmania*, a human protozoan parasite, displays an unconventional DNA replication program in which the timing of DNA replication completion is chromosome size-dependent: larger chromosomes complete replication later then smaller ones. Here we show that both R-loops and RNase H1, a ribonuclease that resolves RNA-DNA hybrids, accumulate in *Leishmania major* chromosomes in a pattern that reflects their replication timing. Furthermore, we demonstrate that such differential organisation of R-loops, RNase H1 and DNA replication timing across the parasite's chromosomes correlates with size-dependent differences in chromatin accessibility, G quadruplex distribution and sequence content. Using conditional gene excision, we show that loss of RNase H1 leads to transient growth perturbation and permanently abrogates the differences in DNA replication timing across chromosomes, as well as altering levels of aneuploidy and increasing chromosome instability in a size-dependent manner. This work provides a link between R-loop homeostasis and DNA replication timing in a eukaryotic parasite and demonstrates that orchestration of DNA replication dictates levels of genome plasticity in *Leishmania*.

In all organisms, DNA replication normally initiates from genomic locations known as origins[1], which are specified in eukaryotes by the binding of the Origin Recognition Complex (ORC)[2]. Eukaryotic cells normally complete replication of their genome during S phase of the cell cycle, before segregating the resulting chromosome copies into offspring cells. To ensure the completion of genome replication in this window of the cell cycle, numerous origins are normally specified and then activated on each eukaryotic chromosome. However, not all origins activate simultaneously at the onset of S phase. Instead, spatial and temporal regulation leads to differential origin activation across the genome as S phase progresses. As a result, the completion of genome duplication occurs in a staggered manner across different chromosome and genome compartments, a phenomenon referred to as 'replication timing'[3–5].

Most of our knowledge about eukaryotic DNA replication timing comes from studies using model organisms, and mainly mammalian cells and yeast, revealing correlations between the orchestration of origin activation and primary and tertiary genomic features. In *Saccharomyces cerevisiae* and related yeasts, origins are conserved DNA elements that bind ORC[6], but replication mapping shows that not all

[1]The University of Glasgow Centre for Parasitology, The Wellcome Centre for Integrative Parasitology, University of Glasgow, School of Infection and Immunity, Sir Graeme Davies Building, 120 University Place, Glasgow G12 8TA, UK. [2]University of Edinburgh, Institute for Immunology and Infection Research, School of Biological Sciences, Edinburgh, UK. [3]Biosciences Institute, Cookson Building, Newcastle University, Framlington Place, Newcastle upon Tyne NE2 4HH, UK. ✉e-mail: jeziel.damasceno@glasgow.ac.uk; richard.mcculloch@glasgow.ac.uk

are activated equally[7]. In all yeasts examined, centromeric origins are activated early[8,9], while subtelomeric origins are activated later[10]. Further domains containing groups of late- or early-activated origins can also be detected[7,11,12], but the basis for such co-ordination is not fully understood, though origin sequence, chromatin and transcription have all been shown to relate to origin activity[13–15]. In the larger genomes of mammals, identification of origins has proved more challenging due to a lack of consistency between datasets that map DNA replication initiation, or localization of ORC[1,16]. In part, such inconsistency may be due to mammalian origins not being sequence-conserved genome features, but could also reflect greater use of stochastic, potentially ORC-independent initiation than is seen in yeast[17–21]. Nonetheless, mammalian replication timing appears to be organized over several scales. The smallest scale is clusters of origins within initiation zones of tens of kb, the earliest activating of which are bounded by efficient origins[22]. Replication timing is not fixed between cell types during mammalian development[23] and cell type-specific 'constant timing regions' have been described, which are large, near megabase-sized regions that comprise smaller (hundreds of kb) replication domains in which origins show coordinated changes in replication timing during development[23,24]. Replication domains display considerable correlation with topologically associated domains revealed by chromatin capture analyses[25,26]. On the largest scale, replication timing is related to nuclear substructuring[27]. Lowly expressed genes are often found towards the periphery of the nucleus in a heterochromatic, lamina-associated domain and are typically late replicating[28]; indeed, increasing the expression of a gene can result in disassociation from the lamina and can advance replication in S-phase for the gene and surrounding sequence[29]. At least some of the genome is associated with the nucleolus and is also characterized by heterochromatin, low levels of gene expression and late replication[30,31]. In contrast, greater gene expression and gene density, as well as higher GC content, is associated with euchromatin and early replication[24]. Indeed, proximity to nuclear speckles (mRNA splicing factor-enriched nuclear bodies) correlates strongly with expression levels, gene density and early replication[32].

DNA replication timing has important implications for genome stability, composition and, ultimately, evolution[33–35]. In both mammals[36,37] and yeast[38,39], levels of single nucleotide polymorphisms have been shown to increase in later replicating parts of the genome. Conversely, in mammals, amplification events through copy number variation[40] and translocations[41] are more abundant in early replicating genomic regions; indeed, early replicating regions are associated with cancerous translocations and altering the timing of DNA replication changes the frequency of such translocations[42,43]. Finally, replication timing correlates with the density of transposable elements in the human genome[23,44,45]. Despite such evidence of its importance, few *cis* and *trans* determinants of DNA replication timing have been described. Recent work identified DNA sequence elements that dictate early replication timing, subnuclear localization and transcription in mouse embryonic stem cells[46], but whether they are common to other mouse cells or wider eukaryotes is unclear. RIF1 (Rap1 interacting factor 1) defines late replicating domains[10,47,48], both through controlling origin activation[49,50] and defining chromatin architecture[51,52]. In contrast, Fkh1/2 (forkhead transcription factor 1/2) defines early replicating domains in yeast[14,53,54]. Finally, in mammals, very long non-coding RNAs, termed asynchronous replication and autosomal RNAs (ASARs), are encoded across the genome and remain associated with the chromosome region from which they are generated[55,56]. ASARs can be expressed from just one allele (termed monoallelic expression)[57,58] or can show different levels of expression from the two alleles[55], but in both cases are correlated with differential replication timing of the chromosomal locus. Disruption of all ASARs so far tested results in delayed replication and instability of the entire chromosome[55,58,59]. How ASARs act in DNA replication timing determination is unclear, but

their role can be mediated via features of LINE retrotransposons, which are highly abundant in mammalian genomes[60]. Whether ASARs form large or punctuated RNA-DNA hybrids (see below) has not been tested.

In contrast to the advanced understanding in model eukaryotes, we are only beginning to unveil how genome duplication is orchestrated in protozoans, which represent much of eukaryotic diversity[61,62]. *Giardia* and *Tetrahymena* cells are unusual amongst eukaryotes in possessing two nuclei, and DNA replication timing of the two genomes appears uncoordinated[63] or constitutively asynchronous[64], respectively. Replication timing within these genomes has not been assessed. In *Plasmodium*, ChIP-seq reveals abundant localization of ORC across the genome that shows limited correlation with mapped origins during schizogony[65,66], and so it is unclear how an observed increase in number of origins activated as S-phase proceeds occurs[66]. In contrast, in the highly transcribed core of the 11 megabase chromosomes of *Trypanosoma brucei*, mapped origins show clear correspondence with ORC binding[67]. In common with all kinetoplastids, virtually all genes in *T. brucei* are expressed from polycistrons[68,69] and ORC and origins localize to the start and ends of these transcription units, meaning they are unusually widely separated and limited in number in the genome[70]. Nonetheless, origin mapping by Marker Frequency sequencing (MFA-seq, equivalent to sort-seq in yeast)[71] suggests variation in origin activation that is conserved across the life cycle of *T. brucei* and between strains[72]. *T. brucei* centromeres appear to be the earliest replicating origins but what dictates variable timing at other origins is unclear, though the wide spacing of origins in *T. brucei* appears to preclude coordinated activation of proximal origins. In addition, replication timing of the large, variable subtelomeres[73] remains uncertain.

*Leishmania* spp are also parasites belonging to the kinetoplastid grouping but show striking differences in genome stability and DNA replication programming relative to *T. brucei*. *Leishmania* parasites display genome-wide intra- and extra-chromosomal gene copy number variation[74,75] and mosaic aneuploidy[76–78], each providing a means of gene expression control and adaptation. Such genome instability is more pervasive than is seen in *T. brucei*, where aneuploidy appears rare[79] and few episomes have been described[80]. An explanation for *Leishmania*'s remarkable genome plasticity could reside in novelties in DNA replication timing. MFA-seq mapped just a single origin in each *Leishmania* chromosome[81], indicating a difference in replication programming compared to the multiple origins/chromosome see in *T. brucei*. Later refinement of MFA-seq in *Leishmania major* indicated persistent, subtelomere-localized DNA synthesis throughout the cell cycle and revealed chromosome length-related replication timing, with larger chromosomes replicated later than smaller ones[82]. Whether such chromosome size-dependent replication timing results from the use of a single predominant origin in each chromosome, with the result that larger chromosomes take longer to be copied[81], or may relate to use of abundant replication initiation activity not detected by MFA-seq[83], is unclear. Here, we sought to determine how this potentially novel chromosome size-dependent replication timing might arise. We show that the distribution of R-loops, a class of RNA-DNA hybrids that form on double-stranded DNA and extrude a single DNA strand, has a striking correlation with *L. major* chromosome length, and that loss of the enzyme RNase H1 alters both DNA replication timing and genome stability. Thus, our works reveals that RNA-DNA hybrids, which are ubiquitous features of all genomes[84,85], are integral to the programming of DNA replication in *Leishmania* and provide a mechanistic link to adaptive genome plasticity.

## Results

### Genome-wide detection of R-loops in *Leishmania major*

R-loops have been characterized from bacteria to mammals, where they play a variety of roles under both physiological and pathological conditions[86]. For instance, R-loops are linked to targeted genome rearrangements to mediate antibody class-switch recombination in

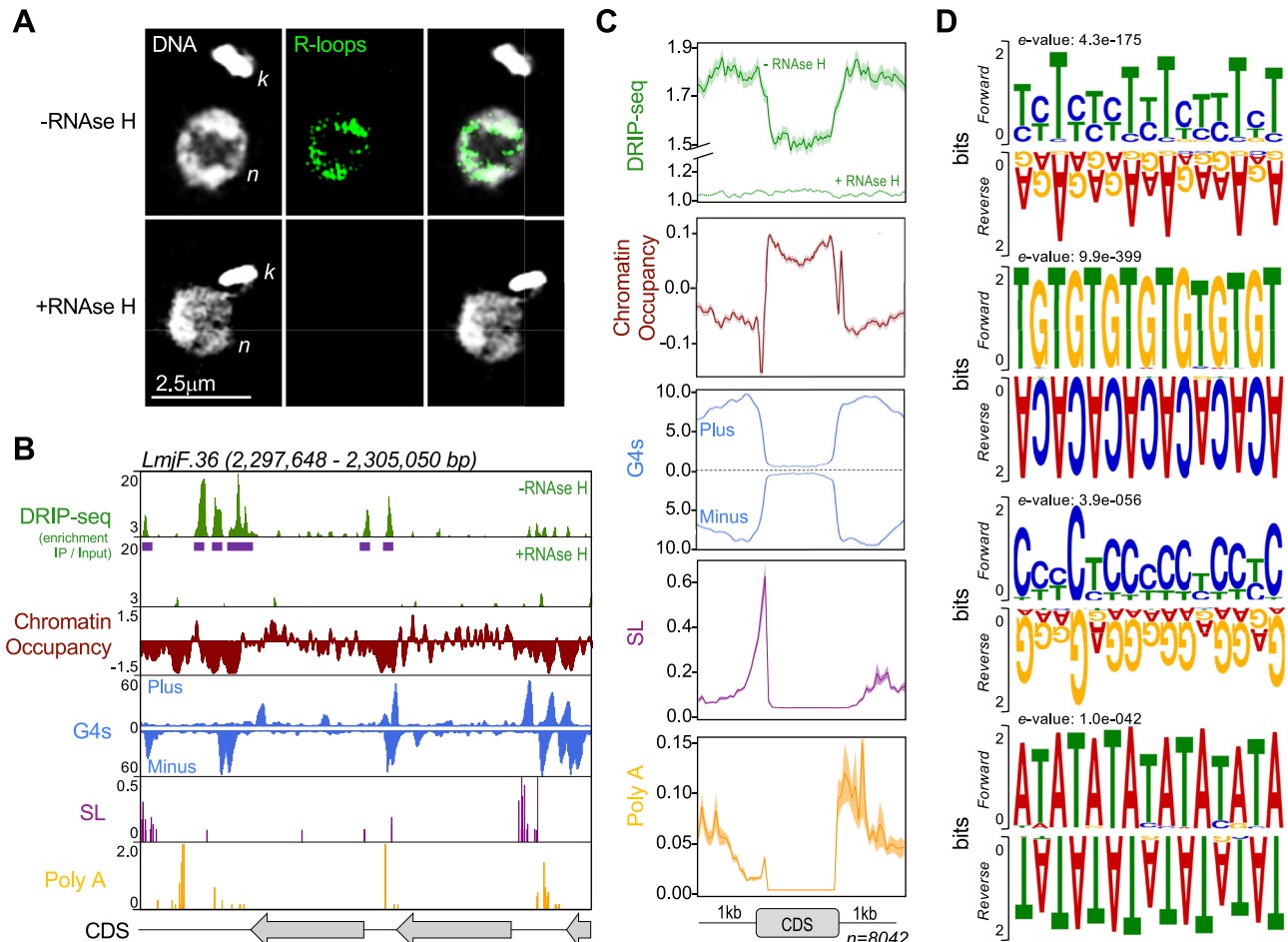

**Fig. 1 | Subcellular localization and genome-wide mapping of R-loops in *L. major*. A** Immunofluorescence analysis to detect R-loops in wild type cells using S9.6 antibody; -RNase H and +RNase H indicate mock or treatment with recombinant RNase HI prior to incubation with antibody, respectively; *n* and *k*, nuclear and kinetoplast DNA, respectively; image is representative of three independent experiments. **B** Snapshot showing DRIP-seq signal relative to the indicated features in a representative genomic region; from top to bottom: track 1 and 2 (green), R-loop enriched regions relative to input material; R-loop peaks are indicated as purple horizontal bars below track 1; track 3 (dark red), chromatin accessibility as determined by MNase-seq; track 4 (blue), G quadruplex structures (G4s) as determined by G4-seq; track 5 (purple), splice leader (SL) acceptor sites as determined by RNA-seq; track 6 (yellow), polyadenylation (Poly A) acceptor sites as determined by RNA-seq; grey arrows at the bottom indicate annotated coding sequences (CDSs); further genomic regions are shown in Supplementary Fig. 3 and 4. **C** Metaplots showing global DRIP-seq signal around CDSs relative to chromatin accessibility, G4 localization, and SL and Poly A sites; lines indicate mean and shaded areas represent SEM. **D** Top enriched DNA sequences motifs found in R-loop peaks, as identified by MEME analysis; *e*-values for each motif are shown on top of each panel; forward and reverse indicate motifs sequences as given by top and bottom strand, respectively, of reference genome.

mammals[87] and can contribute to genome stability by controlling various steps of DNA double strand break repair[88–91]. Moreover, R-loop accumulation is associated with increased local chromatin accessibility[92], activation of gene expression[93,94], and modulation of transcription termination[95,96]. In *T. brucei* R-loops have been implicated in transcription initiation and mRNA maturation[97,98], telomere function[99], and host immune evasion by antigenic variation[100–102]. Reflecting the myriad functions of R-loops, a wide range of proteins have been shown to interact with RNA-DNA hybrids in mammals and *T. brucei*[103,104]. Amongst these are factors that act to remove the RNA-DNA hybrids, including helicases to unwind the structures[105,106] and ribonucleases RNase H1 and RNase H2, which hydrolyse the RNA moiety in R-loops[107]. Unlike in *T. brucei*[97], the distribution of RNA-DNA hybrids has not been examined in *Leishmania*. To assess this, we first examined the subcellular localization of R-loops in *L. major* by performing immunofluorescence using the S9.6 antibody[108] in wild type (WT) cells. Prior to detection with the S9.6 antibody, fixed cells were either left untreated or were treated with recombinant *Escherichia coli* RNase HI. S9.6 signal was mainly seen in the nucleus, where it formed a punctate pattern that was drastically reduced upon pre-treatment with *E. coli*

RNase HI in the presence of MgCl₂ (Fig. 1A). To rule out experimental artifacts due to MgCl₂-mediated RNA degradation by endogenous nucleases or unspecific RNA detection by S9.6[109], we also performed immunofluorescence using the S9.6 antibody after incubating cells in the presence of MgCl₂ alone or after RNase T1 treatment (Supplementary Fig. 1). No significant signal reduction was observed upon MgCl₂ incubation, and while RNase T1 treatment caused a moderate signal reduction, this was significantly less than the signal loss seen after *E. coli* RNase HI treatment (Supplementary Fig. 1). Altogether, these data demonstrate the specificity of the S9.6 antibody for RNA-DNA hybrids in *L. major*.

To provide a genome-wide map of R-loops in *L. major*, we used **D**NA-**R**NA hybrid **i**mmuno**p**recipitation followed by deep **seq**uencing (DRIP-seq) with the S9.6 antibody[110]. Similar to the immunofluorescence analysis, prior to DRIP, genomic DNA was either left untreated or was treated with recombinant *E. coli* RNase HI. Then, DRIP material was subjected to Illumina sequencing (DRIP-seq) and reads mapped to the *L. major* reference genome. While DRIP-seq signal was widely detected in untreated samples, it was drastically reduced at all sites analyzed upon *E. coli* RNase HI pre-treatment, indicating DRIP-seq

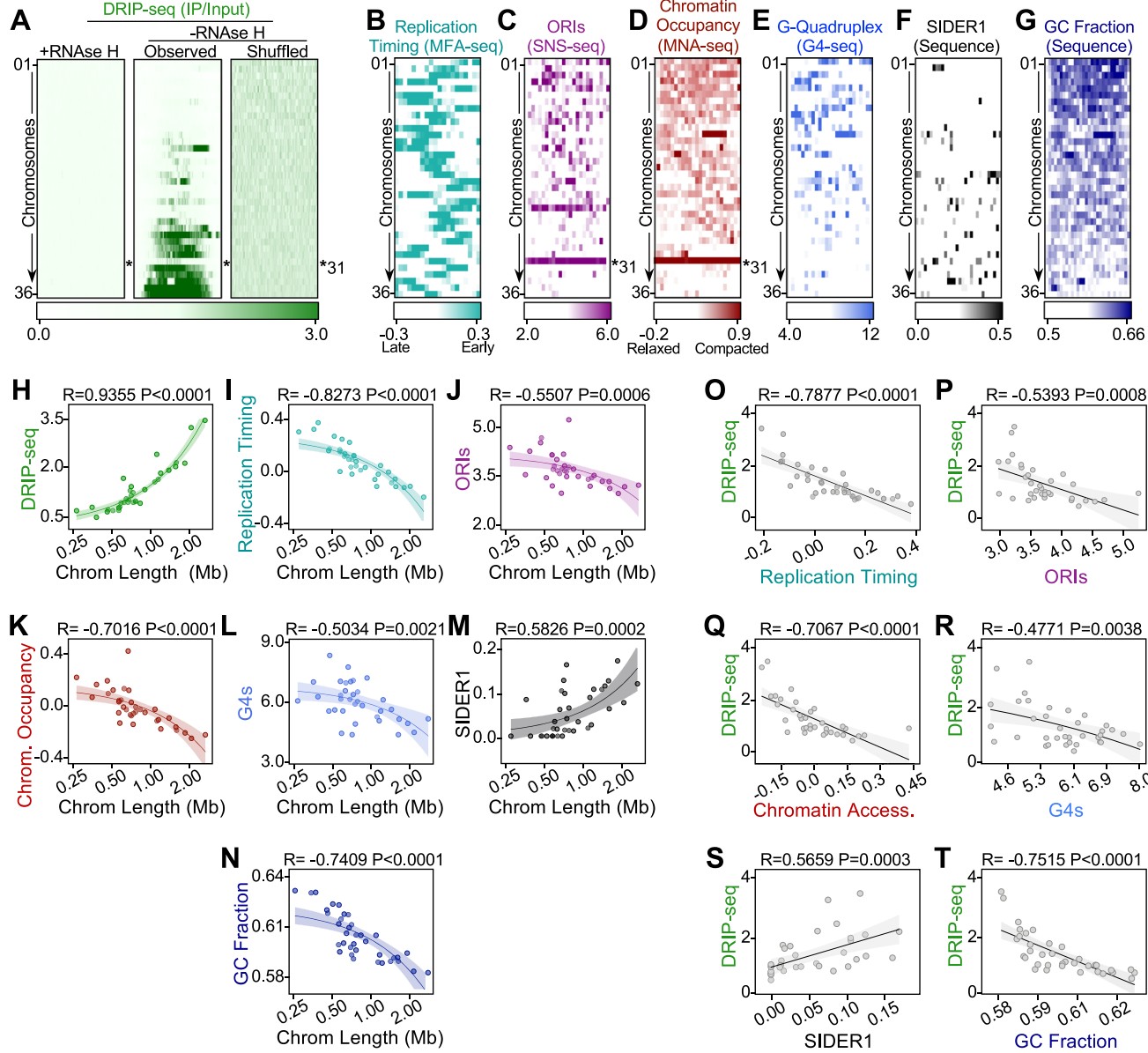

**Fig. 2 | Chromosome-size dependent distribution of R-loops is reflected in a range of further genetic features. A** Colourmap showing distribution of DRIP-seq signal in all 36 *L.major* chromosomes; chromosomes are ordered by size; -RNase H and +RNase H indicate mock or treatment with recombinant RNase HI prior to immunoprecipitation, respectively; shuffled, indicates DRIP-seq signal plotted after R-loops peaks were randomly distributed throughout the genome; an independent experiment is shown in Supplementary Fig. 5. **B–G** Colourmaps showing distribution patterns of DNA replication timing predicted by MFA-seq, putative origins of DNA replication (ORIs) predicted by SNS-seq, chromatin accessibility determined by MNase-seq, G-quadruplexes (G4s) density determined by G4-seq, distribution of directed and inverted <u>s</u>hort <u>i</u>nterspersed <u>de</u>generate <u>r</u>etroposons 1 (SIDER1) and GC

fraction, respectively; chromosome 31, which does not follow the pattern of all other chromsomes for (**A**, **C** and **D**), is indicated. **H–N** Simple linear regression analysis showing correlation between chromosome size and chromosome-averaged signals of DRIP-seq, DNA replication timing, ORIs, chromatin accessibility, G4s, SIDER1 sequences and GC fraction, respectively. **O–T** Simple linear regression analysis showing correlation between averaged DRIP-seq signal at each chromosome and averaged signals of DNA replication timing, ORIs, chromatin accessibility, G4s and SIDER sequences, respectively. In panels **H** to **T**, **R** and **P** values are indicated at the top of each panel; circles indicate mean, lines indicate the best fit and shaded areas represent 95% CI. Source data are provided as a Source Data file.

detects locations of RNA-DNA hybrids (Fig. 1B, C; see also Fig. 2A and Supplementary Figs. 3, 4 and 5). To understand the localization of RNA-DNA hybrids, we compared the distribution of DRIP-seq signal to a number of genome features, revealing that R-loops display pronounced accumulation at regions between coding sequences (CDSs) in polycistronic transcription units (Fig. 1B), where nascent transcripts are abundant (Supplementary Fig. 2A, B), chromatin occupancy is lower (as determined by MNase-seq)[83] and G-quadruplex (G4)[111] occurrence is higher (Fig. 1B, C, Supplementary Fig. 3). At a finer level, R-loop accumulation in inter-CDS regions overlapped with splice

leader (SL) and polyadenylation (Poly A) acceptor sites (Fig. 1B, C). To examine this association further, we identified 12,219 DRIP-seq peaks across the *L. major* genome and used MEME[112] to identify DNA sequences that were enriched, which revealed two motifs composed of polypyrimidines ($TC_n$ and $CCT_n$) and two other motifs composed of either TA or TG repeats (Fig. 1D). RNA-DNA hybrid formation is favoured at purine-rich sequences in other eukaryotes[113]. A significant proportion of the R-loop-associated $TC_n$ and $CCT_n$ motifs in *L. major* are on the antisense DNA strand within a transcription unit (Supplementary Fig. 2C), suggesting that R-loops that form in these regions are

composed of RNA molecules corresponding to the reverse complement genome sequence, and are also therefore found at purine-rich sequence. R-loops in *T. brucei* are also enriched in polypyrimidine-containing regions[97], suggesting the existence of trypanosomatid-specific mechanisms for R-loop stabilization or resolution when they form within transcription units in the sense DNA strand. Since polypyrimidine tracts and their binding proteins have been shown to act in *trans*-splicing and polyadenylation across kinetoplastids[114–116], these data indicate an potentially kinetoplastid-wide association between R-loops and RNA processing events to generate mature mRNAs during multigene transcription. In addition, we also found sequences associated with G4 formation to be enriched at DRIP-seq peaks (Supplementary Fig. 2D), consistent with R-loop and G4 co-localization. DRIP-seq signal was also found in the vicinity of tRNA genes (Supplementary Fig. 4), as seen in yeast[117].

### The global distribution of R-loops correlates with *Leishmania* chromosome size-related DNA replication timing

In analyzing R-loop distribution we noted that, remarkably, DRIP-seq density displayed a significant correlation with *L. major* chromosome length (Fig. 2A, H; independent replicate shown in Supplementary Fig. 5A), with R-loops becoming more abundant as chromosome size increased, though it is notable that chromosome 31 did follow this pattern. Moreover, DRIP-seq signal increased towards the centre of each chromosome, with concomitant depletion from the sub-telomeres, indicating a gradient of R-loop levels from the chromosome cores to their ends (Supplementary Fig. 5B). Importantly, these patterns were lost in samples pre-treated with *E. coli* RNase HI and upon randomization of DRIP-seq signal, ruling out intrinsic bias from the immunoprecipitation or mapping strategy (Fig. 2A and Supplementary Fig. 5).

It could be argued that chromosome length is unlikely to be the determinant of such an unusual R-loop distribution, but that it reflects asymmetric activities across the genome. Therefore, we tested how R-loop distribution correlates with known features of the *L. major* DNA replication programme. Our previous work[82] using MFA-seq showed larger *L. major* chromosomes are, on average, replicated later when compared to smaller chromosomes (Fig. 2B, I), with R-loop density and chromosome replication timing therefore showing a significant anti-correlation (Fig. 2O). Lombrana et al.[83] used SNS-seq (short-nascent strand sequencing) to map thousands of predicted DNA replication origins in *L. major*. In light of our R-loop mapping, we reanalyzed the SNS-seq data and again found a significant chromosome size-dependence, with larger chromosomes presenting a lower density of predicted origins than smaller ones (Fig. 1C, J). As a result, SNS-seq origin density significantly negatively correlates with the average R-loops level of each chromosome (Fig. 2P). This analysis indicates that chromosome size-dependent replication timing is reflected in a similarly skewed distribution of R-loops and SNS-seq predicted DNA replication initiation events.

### Global R-loop distribution correlates with chromosome size-related chromatin accessibility, G-quadruplex levels and DNA sequence content

We next asked if the global DRIP-seq pattern is reflected in wider features of *L. major* chromosome activity and sequence composition. First, we compared both nascent[118] and mRNA levels among chromosomes: only a modest correlation between mRNA levels and chromosome length was seen, and nascent transcripts level did not significantly change with chromosome size (Supplementary Fig. 7). Next, we looked at chromatin occupancy by remapping available MNase-seq[83] data across the genome, which revealed higher occupancy as chromosomes reduce in size (Fig. 2D, K), meaning there is a significant anticorrelation with DRIP-seq signal (Fig. 2Q). This finding indicates that larger chromosomes with higher R-loop levels present

relatively more relaxed chromatin when compared to smaller ones, suggesting that chromatin accessibility is not just a determinant of local R-loop accumulation (Fig. 1B, C), but of global distribution as well. Re-analysis of G4-seq data revealed a similar and significant anticorrelation with DRIP-seq, with higher G4s level in the smaller chromosomes than the larger (Fig. 2E, L, R). Thus, despite G4 and R-loop densities positively correlating with each other in specific, local genomic areas (Fig. 1C), they are negatively correlated genome-wide.

Remarkably, we next observed that R-loop, chromatin and G4 levels are reflected in size-dependent variation in *L. major* chromosome sequence content. First, we examined the distribution of direct and inverted repeat DNA sequences, including *s*hort *i*nterspersed *de*generate *r*etroposons 1 and 2 (SIDER1 and SIDER2, respectively), which are related to gene copy number variation (CNV)[119] and regulation of gene expression[120,121] in *Leishmania*. Both non-SIDER and SIDER2 repeats appear to be present in homogenous levels across chromosomes (Supplementary Fig. 6A, B, E and F). Also, we found no evidence that GC or AT skew displays any correlation with chromosome length (Supplementary Fig. 6C, D, G and H). However, SIDER1 repeats were found to be enriched in larger chromosomes when compared to smaller ones (Fig. 2F, M), significantly correlating with DRIP-seq levels (Fig. 2S). Because retrotransposition can give rise to R-loops in yeast[117], we plotted DRIP-seq signal around SIDER1, SIDER2 and non-SIDER repeats. Despite SIDER1 repeats being more abundant in chromosomes with higher R-loops density (Fig. 2S), no localized enrichment of DRIP-seq signal overlapping these regions was detected. This suggests SIDER1 are not significant sources of R-loop generation in the larger chromosomes in this parasite (Supplementary Fig. 8). Finally, we also observed that GC content decreased as chromosome size increased (Fig. 2G, N), displaying a significant anticorrelation with DRIP-seq levels (Fig. 2T).

Altogether, these observations indicate a previously undetected asymmetry in many aspects of chromosome content and function in *L. major*: patterns of R-loop accumulation among chromosomes can be correlated not only with chromatin accessibility and the unconventional DNA replication timing programme of the parasite, but also with the evolution of genome architecture, as reflected in sequence content biases among chromosomes.

### *L. major* RNase H1 localizes to the nucleus and associates with strand switch regions

To test for potential functional consequences of R-loop distribution in *L. major*, we next focused on the factors involved in RNA-DNA hybrid resolution. Two ribonuclease H enzymes are known to participate in the resolution of R-loops in eukaryotes[107], including *T. brucei*[98,100]. In *L. major* one predicted subunit of trimeric RNase H2 has been reported to act in the mitochondrion[122], and so we focused on RNase H1, which has not been studied in *Leishmania* but provides nuclear functions in *T. brucei*[100]. We used CRISPR/Cas9 to flank the endogenous *RNase H1* ORF with *loxP* sites (flox), allowing the gene to be deleted by rapamycin-mediated induction of DiCre activity (Supplementary Fig. 9A). In addition, the *RNase H1* ORF was translationally fused with 6 copies of the HA epitope at the C-terminus to allow us to monitor protein levels and location before and after gene excision. PCR showed that all copies of *RNase H1* were floxed and HA-tagged after a single round of transformation (Supplementary Fig. 9B); the resulting cell line is hereafter referred to as *RNase H1-HA^Flox^*. Simultaneous addition of *loxP* sites and the HA tag did not result in any significant growth defect in *RNAse H1-HA^Flox^* cells compared with WT (Supplementary Fig. 9C).

To investigate the subcellular localization of RNase H1 in *L. major* and its potential relationship with DNA replication, we employed immunofluorescence analysis with *RNAse H1-HA^Flox^* cells pulsed with EdU to label cells in S phase. These experiments indicated that RNase H1-HA primarily localized to the nucleus, with minimal colocalization with nascent DNA (Fig. 3A, B) and higher abundance in non-replicating

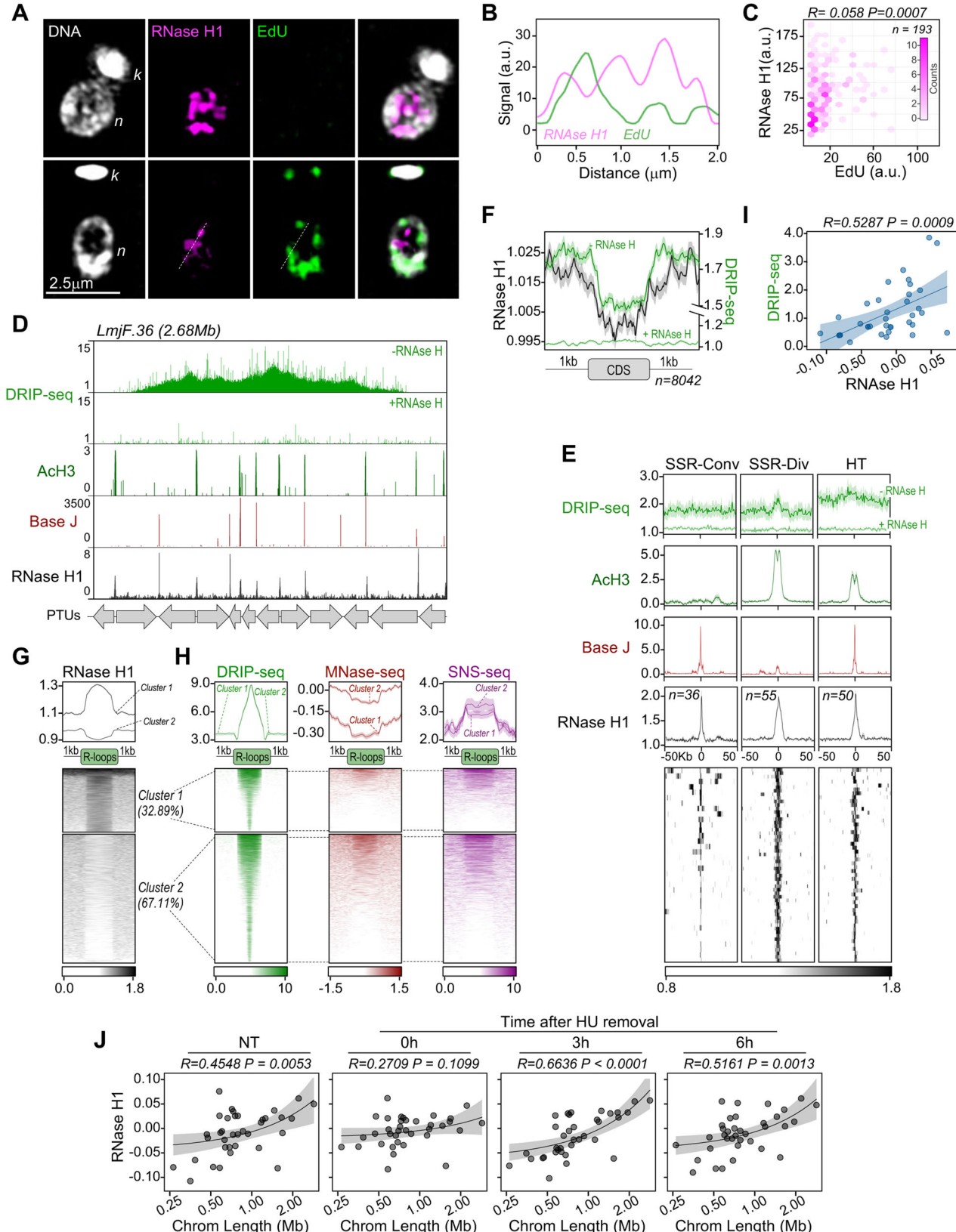

cells (Fig. 3C). Based on this, we conclude that RNase H1 provides nuclear functions in *L. major* and does not appear to be temporally or spatially linked to replicative DNA synthesis.

Next, we performed chromatin immunoprecipitation followed by deep sequencing (ChIP-seq) with anti-HA antiserum to characterize the genome-wide binding profile of RNase H1-HA. Visual inspection

indicted that that RNase H1-HA was most strongly enriched at the boundaries of the polycistronic transcription units (PTUs), termed strand switch regions (SSRs), co-localizing with acetylated histone H3 (AcH3)[123] and β-D-glucosyl-hydroxymethyluracil (Base J)[124], markers of transcription initiation and termination sites, respectively (Fig. 3D, E). Furthermore, visual comparison between RNase H1-HA ChIP-seq and

**Fig. 3 | Subcellular localization and genome wide mapping of RNase H1 in *L. major*. A** Immunofluorescence analysis to detect RNase H1-HA using anti-HA antibody; cells undergoing DNA replication are shown by EdU signal; *n* and *k* indicate nuclear and kinetoplast DNA, respectively; image is representative of two independent experiments. **B** Line scan, plotting the RNase H1-HA and EdU signal intensity values across the dotted white line in (**A**). **C** RNase H1-HA versus EdU signal intensity plotted as a 2D density plot using hexagonal bins; *R* and *P* values for linear regression analysis is shown at the top. **D** Representative snapshot of RNase H1-HA ChIP-seq; from top to bottom: track 1 and 2 (green), DRIP-seq signal where -RNase H and +RNase H indicate mock or treatment with recombinant RNase HI prior to immunoprecipitation, respectively; track 3 (dark green), enriched regions of acetylated Histone H3 (AcH3); track 4 (dark red), β-D-glucosyl-hydroxymethyluracil (Base J) enriched regions; track 5 (dark grey), RNase H1-HA enriched regions relative to input material; grey arrows at the bottom indicate the position and orientation of polycistronic transcription units (PTUs). **E** Metaplots (top) and colourmap (bottom) showing RNase H1-HA ChIP-seq signal around convergent, divergent and head-to-tail strand switch regions (SSR-Conv, SSR-Div and HT, respectively); metaplots for DRIP-seq, AcH3 and Base J are also shown; in metaplots above colourmaps, lines and shaded areas represent mean and SEM, respectively. **F** Metaplot showing global RNase H1-HA ChIP-seq signal (dark gray) around CDSs compared with DRIP-seq signal (green); lines indicate mean and shaded areas represent SEM. **G** Metaplots (top) and colourmap (bottom) showing RNase H1-HA ChIP-seq signal around DRIP-seq peaks; regions were grouped using *k-means* clustering; percentages indicate the proportion of peaks in each cluster. **H** DRIP-seq, MNase-seq and SNS-seq signals were plotted around DRIP-seq peaks grouped in clusters 1 and 2 from (**G**) and represented as metaplots (top) and colourmaps (bottom). In metaplots above colourmaps (in **G** and **H**), lines and shaded areas represent mean and SEM, respectively. **I** Simple linear regression analysis showing correlation between averaged signals of RNase H1-HA ChIP-seq and DRIP-seq at each chromosome. **J** Simple linear regression analysis showing correlation between chromosomes length and averaged signals of RNase H1-HA ChIP-seq at each chromosome in unperturbed exponentially growing cells (NT) and after release from synchronization with hydroxyurea (HU); cell cycle progression analysis upon HU synchronization is shown in Supplementary Fig. 14A. In panels **I** and **J**, *R* and *P* values are indicated at the top of each panel, circles indicate mean, lines indicate the best fit and shaded areas represent 95% CI. Source data are provided as a Source Data file.

DRIP-seq signals indicated considerable overlap between RNase H1-HA and R-loop peaks (Fig. 3D), an observation supported by metaplots that showed elevated DRIP-seq signal around SSRs, where RNASE H1-HA accumulated (Fig. 3E), and overlap of RNase H1-HA ChIP-seq and DRIP-seq signal at inter-CDS regions within PTUs (Fig. 3F). Importantly, however, such co-localization was not uniform, since examination of RNase H1-HA ChIP-seq signal around DRIP-seq peaks suggests that ~33% of R-loop regions are RNase H1-bound, while the remaining ~67% are RNase H1-free (Fig. 3G). The explanation for the difference between these two classes of R-loops, which present similar average levels of DRIP-seq and SNS-seq signals, appears to lie in chromatin accessibility, since RNase H1-bound R-loops are located at genomic regions with lower average MNase-seq signal (Fig. 3H).

Given the chromosome-size dependent distribution of R-loops (Fig. 2), we next asked if RNase H1 might accumulate across the genome in a similar way. Quantification of the ChIP-seq signal indicated that RNase H1-HA does not associate equally to all chromosomes, since we found a significant correlation between average RNase H1-HA ChIP-seq and DRIP-seq signals for each chromosome (Fig. 3I), and also between RNase H1-HA ChIP-seq signal and chromosome length in asynchronous cultures (Fig. 3J, first panel). Moreover, this correlation was lost when *L. major* cells were arrested in G1 by hydroxyurea treatment but increased as cells were released from G1 arrest and synchronously navigated through S-phase and G2/M (0, 3 and 6 hrs; Fig. 3J, second to fourth panels; cell cycle progression upon synchronization with hydroxyurea is shown in Supplementary Fig. 14A). These analyses suggest two things. First, the global distribution of *L. major* RNase H1 mirrors that of the R-loops that it acts to dissolve, with greater accumulation in larger chromosomes. Second, despite no evidence of overlap between RNase H1-HA signal and DNA synthesis detected by EdU, chromosome-size dependent accumulation of RNase H1 is cell cycle-dependent and is more marked during S-phase to G2/M transition, perhaps reflecting *Leishmania* chromosome size-dependent DNA replication timing.

## Loss of *L. major* RNase H1 results in a transient growth defect and R-loop accumulation under DNA replication stress

To evaluate the consequences of RNase H1 loss, conditional knockout (*KO*) of the floxed *RNase H1-HA* gene was induced by rapamycin-mediated DiCre activation in logarithmically growing cultures of *L. major* (Fig. 4A). DiCre-mediated loss of the protein was confirmed by western blotting, where signal for RNase H1-HA was no longer detectable after 48 h of the second round of *KO* induction (Fig. 4B, Supplementary Fig. 10). Genome-wide DNA and RNA sequencing showed loss of the *RNase H1* gene and its RNA upon induction of DiCre activity, confirming precise DiCre excision (Supplementary Fig. 11).

The timing of RNase H1-HA loss coincided with marked slowing in growth of the induced cells compared with uninduced, an effect that continued for around 20 days (until passage 5, approximately 40 population doublings for WT cells) but was followed by recovery, with the DiCre-induced cells' growth becoming indistinguishable from uninduced cells by passage 6 (Fig. 4C). Importantly, PCR analysis showed that unexcised copies of *RNase H1-HA* were undetectable in induced cells throughout the course of the experiment, ruling out the possibility that reversion of the growth defect was due to restoration of the *RNase H1* gene or outgrowth of cells with unexcised floxed gene (Fig. 4D). These data suggest that loss of RNase H1 leads to deleterious effects in *L. major* promastigotes in the short-term, from which cells recover in the long-term. The data also suggest that loss of *RNase H1* is not lethal. To test this prediction, we subcloned the DiCre induced cells after passage 2 (Fig. 4C, Supplementary Fig. 10). A clonal cell line (Supplementary Fig. 9D and 11), hereafter referred as *KO*, was recovered that showed no evidence for the presence of *RNase H1* gene copies or mRNA (Supplementary Fig. 9D and 11) and did not present any detectable growth defect compared with WT cells (Fig. 4E). Altogether, these data indicate that loss of RNase H1 is transiently detrimental to *L. major* proliferation, but the parasites can adapt and recover fecundity. In addition, the *KO* cells provide a means of comparing short- and long-term effects resulting from *RNase H1* loss.

Next, we asked if RNase H1 loss leads to accumulation of R-loops. For this, we performed immunofluorescence analysis using the S9.6 antibody, comparing the levels of nuclear R-loops signal in uninduced cells relative to when *RNase H1* excision caused growth impairment and also in the *KO* cells. Surprisingly, we did not observe any significant difference in S9.6 signal in any of these conditions (Fig. 4F, G). However, when cell cycle progression at G1/S was blocked via hydroxyurea treatment, we observed a significant accumulation of R-loops upon both short- and long-term *RNase H1* loss (Fig. 4F, G). Importantly, this effect appears specific to DNA replication stress resulting from hydroxyurea exposure, since treatment with camptothecin (blocks DNA replication by inhibiting the activity of topoisomerases)[125], actinomycin D (blocks transcription elongation)[126], flavopiridol (blocks cells cycle at G2/M by inhibiting CRK3 Cyclin-Dependent Kinase)[127] or AB1 (blocks cells cycle at G2/M by inhibiting kinetochore assembly)[128] did not result in significant change in nuclear R-loop signal (Supplementary Fig. 12). These data suggest RNase H1 is dispensable for R-loop resolution either because R-loops do not accumulate under normal growth conditions, or because of functional compensation by other activities, such as RNase H2. Alternatively, it is possible that R-loops do accumulate upon loss of RNase H1 under unstressed conditions, but only in discrete genomic locations, thus escaping detection via immunofluorescence. Nonetheless, the pronounced change in R-loop

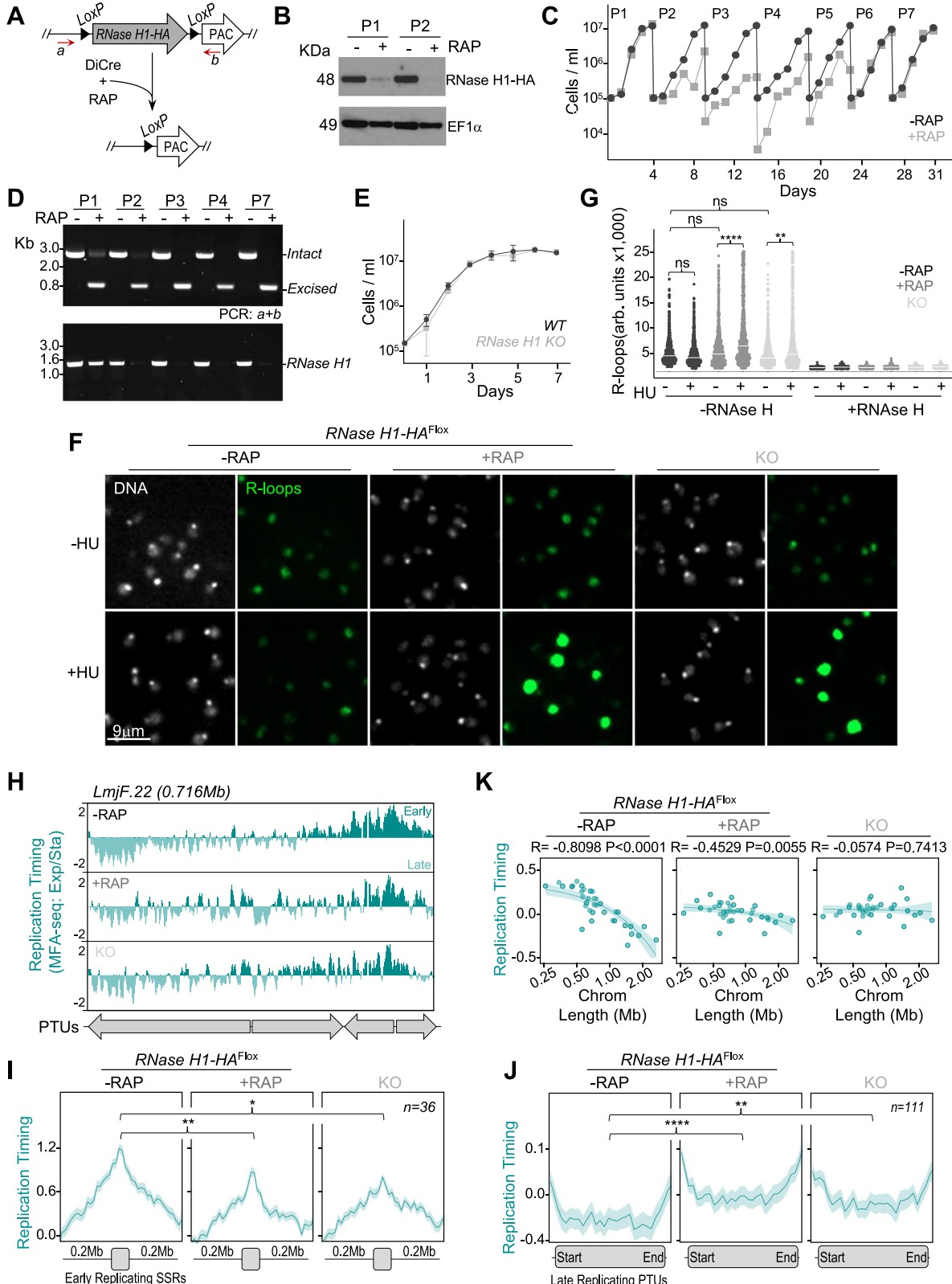

**Loss of RNase H1 abrogates *Leishmania* chromosome size-dependent DNA replication timing**

Since the distribution of R-loops and RNase H1 has parallels with chromosome size-dependent timing of *L. major* DNA replication (Fig. 2

and Fig. 3), we reasoned that the ribonuclease could be a hitherto undetected player that directs the DNA replication programme of the parasite. To test this prediction, we performed MFA-seq analysis in uninduced and induced *RNase H1-HA^Flox* cells at passage 2, as well as in the *KO* cells, allowing us to compare the short- and long-term effects of *RNase H1* loss. The MFA-seq profile in uninduced *RNase H1-HA^Flox* cells corresponded with previous reports[81], with a single pronounced peak

levels in the absence of RNase H1 after hydroxyurea treatment lends further weight to a cell cycle-dependent role for RNase H1.

**Fig. 4 | Effects of *RNase H1* loss on growth, R-loop accumulation and DNA replication timing. A** Schematic representation of the DiCre-mediated *RNase H1* gene deletion strategy; CRISPR-Cas9 was used to flank the *RNase H1* ORF with *LoxP* sites and fuse it with an HA tag (*RNase H1-HA^Flox^*); rapamycin-mediated activation of DiCre was used to catalyze excision of *RNase H1-HA^Flox^*; refer to Supplementary Fig. 10 for the rapamycin induction strategy; *a* and *b*, annealing position of primers used in (**D**). **B** Western blotting analysis of whole cell extracts from *RNase H1-HA^Flox^* cells -48 h after growth in the absence (−RAP) or in the presence (+RAP) of rapamycin at passages 1 and 2 (P1 and P2, as shown in (**C**)); extracts were probed with anti-HA antibody and anti-EF1α was used as loading control. **C** Growth profile of the *RNase H1-HA^Flox^* cell line cultivated in the absence (−RAP, black) or presence (+RAP, grey) of rapamycin; cells were seeded at -10^5 cells.mL^−1 at day 0 and diluted back to that density every 4–5 days for seven passages (P1 to P7); cell density was assessed every 24 h in two independent experiments. **D** PCR analysis of genomic DNA extracted from *RNase H1-HA^Flox^* cells -48 h in the indicated passages, after growth in the absence (−RAP) or in the presence (+RAP) of rapamycin; annealing positions for primers *a* and *b* are shown in (**A**); image is representative of two independent experiments. **E** Growth profile of a clonal *RNase H1 KO* cell line, selected after DiCre-mediated *RNase H1* gene deletion compared to wild type (WT) cells; cell density was assessed every 24 h and is represented as the mean from four independent experiments; error bars indicate SEM. **F** Immunofluorescence analysis using S9.6

antibody to detect R-loops with (+HU) or without (−HU) 5 mM hydroxyurea treatment for 6 h. **G** Quantification of R-loops levels detected via immunofluorescence using S6.9 antibody in the indicated conditions, represented as arbitrary units (arb. units); −RNase H and +RNase H indicate mock or treatment with recombinant RNAse HI prior to incubation with antibody, respectively; quantification is representative of three independent experiments; (****),(**) and ns: $p < 0.0001$, $p = 0.0089$ and not significant, respectively, as determined by Kruskal−Wallis test (one-way ANOVA) using Dunn's test for multiple comparison correction. **H** Representative snapshot showing DNA replication timing on the entire chromosome 22, as determined by MFA-seq using exponentially growing cells normalized with stationary cells; positive and negative values indicate early and late replicating regions, respectively; grey arrows at the bottom indicate the position and orientation of polycistronic transcription units (PTUs). **I, J** Metaplots showing global MFA-seq signal in early and late replicating regions, respectively; lines indicate mean and shaded areas represent SEM; (****),(**), (*) and ns: $p < 0.0001$, $p = 0.0054$, $p = 0.0188$ and not significant, respectively, as determined by Kruskal−Wallis test (one-way ANOVA) using Dunn's test for multiple comparison correction. **K** Simple linear regression analysis showing correlation between chromosome size and averaged MFA-seq signal; R and *P* values are indicated at the top of each panel, circles indicate mean, lines indicate the best fit and shaded areas represent 95% CI. Source data are provided as a Source Data file.

in each chromosome that overlapped with an SSR (Fig. 4H). However, when comparing the MFA-seq profile around the early-replicating SSR of each chromosome and in late-replicating PTUs (Supplementary Fig. 13), both the induced *RNase H1-HA^Flox^* cells and the *KO* cells displayed a striking change: MFA-seq signal around every early-replicating SSR was decreased (Fig. 4H, I) and, concomitantly, increased MFA-seq signal was observed across late replicating PTUs in both the induced *RNase H1-HA^Flox^* cells and the *KO* cells (Fig. 4H, J). This result suggests that RNase H1 activity is required for maintenance of the DNA replication programme in *L. major*.

To further test this, we performed linear regression analysis of averaged MFA-seq signal for each chromosome against its length and compared the resulting profile from uninduced cells with those from induced and from *KO* cells. Strikingly, the correlation between replication timing and chromosome length was dramatically weaker in induced cells and absent in the *KO* cells (Fig. 4K). Interestingly, this change in DNA replication timing upon loss of RNase H1 was not accompanied by pronounced cell cycle progression defects (Supplementary Fig. 14A), or by alterations in the proportion of cells in S phase (Supplementary Fig. 14B). Altogether, these data suggest that RNase H1 has a pivotal role in the DNA replication programme of *L. major* and that reversion of a growth defect seen shortly after RNase H1 loss correlates with abrogation of the major DNA replication timing dynamic, which is chromosome length-dependent.

## Loss of RNase H1 leads to genome instability

Because DNA replication is key for genome maintenance and transmission, we set out to investigate if the shift in the replication program upon loss of RNase H1 might affect genome stability by performing short-read Illumina whole genome sequencing to test for levels of CNV events during growth. We did this by calculating fold change in normalized read numbers from uninduced and induced *RNase H1-HA^Flox^* cells upon short-term cultivation (three passages), as well as from long-term cultivation (*KO* cell line) across the entirety of each chromosome (Fig. 5A). This genome-wide analysis revealed that short term cultivation of induced *RNase H1-HA^Flox^* cells displayed a modest increase in CNV relative to uninduced cells, and that this change was more prominent in the cores of chromosomes and increased to a greater extent in the larger chromosomes compared with the small ones (Fig. 5B). CNV events more clearly reflected R-loop abundance when comparing the *KO* cells to uninduced *RNase H1-HA^Flox^* cells (Fig. 5B, C), indicating that the CNV change was greater after prolonged cultivation of RNase

H1 *KO* cells. In addition, this analysis revealed apparent loss of DNA sequences in the *KO* cells at subtelomeres, and at the repetitive ribosomal- and SL-RNA-encoding *loci* (Fig. 5E), which in *L. major* and other organisms are a pronounced region of R-loop accumulation and where loss of RNase H affects their stability[97,129–131]. To test if CNV occurrence in these regions correlates with RNase H1 action on R-loops, we next compared the CNV and DRIP-seq data with RNase H1-HA ChIP-seq. Although rRNA- and SL-RNA loci showed pronounced accumulation of both RNaseH1 and R-loops, no such pronounced R-loop enrichment was observed at the subtelomeres (Fig. 5E). CNV also clearly correlated with wider regions of DRIP-seq signal enrichment in both uninduced and induced cells, and the extent of this variation was greater at those loci that were bound by RNase H1 (Fig. 5D). In *KO* cells, however, more pronounced CNV was seen around RNase H1-free regions than at RNase H1-bound regions. These data indicate that R-loops acted upon by RNase H1 are pronounced regions of CNV both globally and locally, though loss of the endonuclease may result in adaptation that is reflected in a changed pattern of such variation.

Genome sequencing also revealed both dramatic (chromosomes 05 and 12) and milder (chromosomes 04, 14 and 31) whole-chromosome CNV in the *KO* cells, as well as modest but significant whole-chromosome CNV during the limited growth period after DiCre induction (chromosomes 04, 05, 14, 20 and 31; Fig. 5B, F). The median of normalized coverage for chromosomes 04, 12, 14 and 20 from the starting, uninduced cultures indicate they were nearly euploid, with 2.1, 3.1, 2.1 and 3.0 copies, respectively. Upon either short- or long-term growth after RNase H1 loss their copy number dropped to 1.84, 2.16, 2.95 and 2.56, respectively. On the other hand, chromosomes 05 and 31, which were aneuploidy in the starting uninduced cells with 2.7 and 5.2 median copies, respectively, showed a drop to 2.24 and 4.86 during short-term cultivation, respectively.

Altogether, these data suggest that reprograming of the DNA replication landscape upon RNase H1 loss leads to pervasive CNV at R-loops as well as alterations in ploidy control, with such effects being more severe after long-term growth. The loss of subtelomere, ribosomal and SL sequences suggests that RNase H1 is required for the maintenance of repetitive DNA, possibly by preventing R-loops accumulation at DNA replication stress-prone regions. It is also possible that at least some of the stable changes in karyotype may explain reversion of the short-term growth defect after RNase H1 loss, consistent with the idea that aneuploidy is an adaptive strategy relying on polyclonal selection of pre-existing karyotypes[132,133].

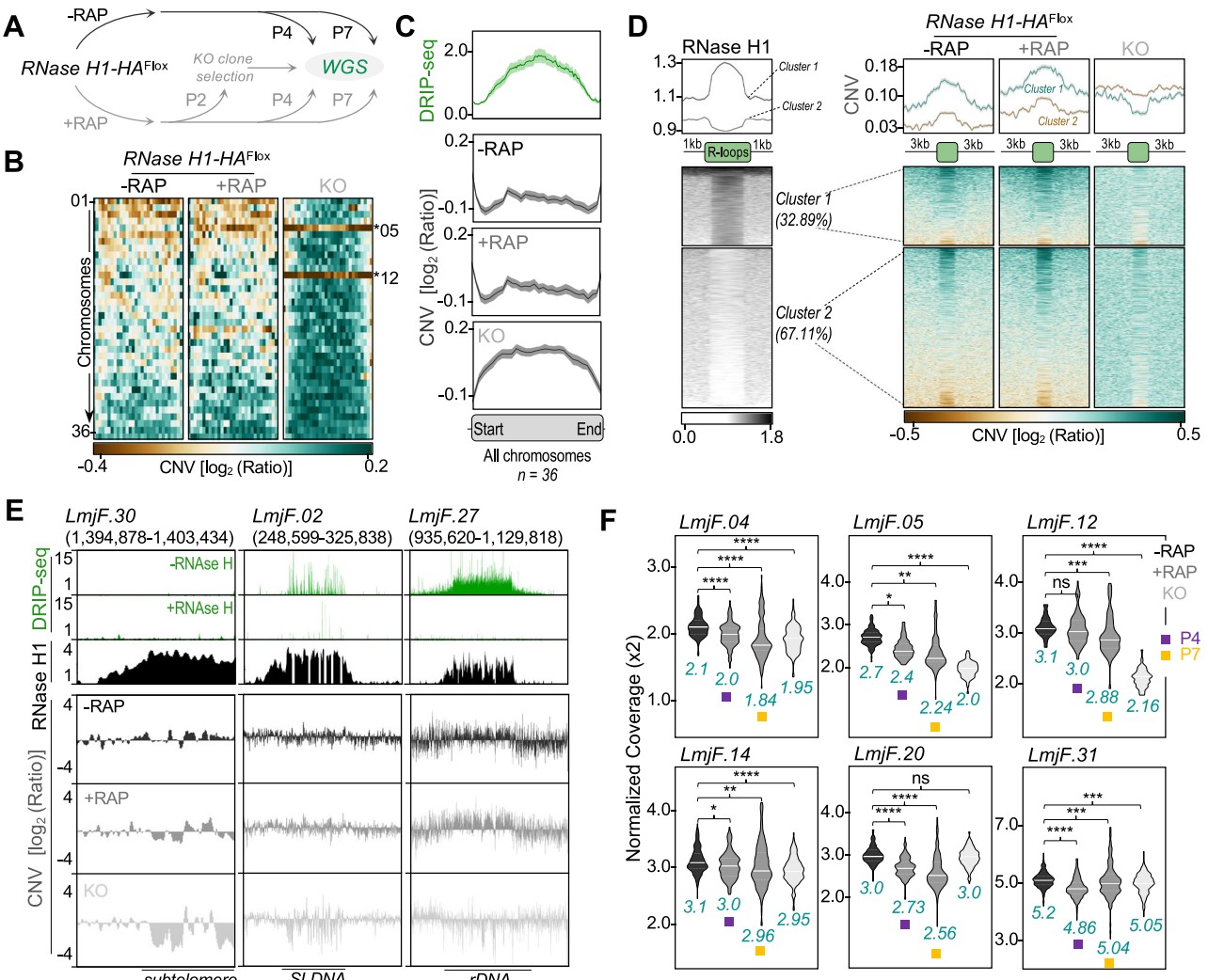

**Fig. 5 | Analysis of CNV events upon DiCre-mediated *RNase H1* gene deletion.** **A** Schematic representation showing time points from which cells were collected for whole genome sequencing (WGS). **B** Colourmap showing genome-wide relative copy number variation (CNV) analysis; chromosomes are ordered by size from top to bottom; CNV in *RNase H1-HA$^{Flox}$* cells is expressed as log$_2$[ratio(normalized reads from P7/normalized reads from P4)] for either −RAP or +RAP conditions; CNV in *KO* cell line is expressed as log$_2$ [ratio(normalized reads from KO/ normalized reads from P4 -RAP)]; chromosomes 05 and 12, showing decreased copy number, are indicated. **C** Metaplots showing averaged CNV profiles across all chromosomes (grey); averaged DRIP-seq profiles across all chromosomes from 2A is also shown at the top (green). **D** Metaplots (top) and colourmap (bottom) showing relative CNV profiles around DRIP-seq peaks upon *k-means* clustering from 3G. In metaplots from C and D, lines indicate mean and shaded areas represent SEM. **E** Relative CNV analysis at the indicated *loci*; CNV is expressed as in (**B**); DRIP-seq (green) and RNase H1-HA ChIP-seq (black) signals are also shown at the top. **F** Absolute chromosome CNV analysis for the indicated chromosomes in the indicated conditions and passages; violin plots represent the distribution of normalized read counts relative to the haploid genome content; (****),(***) and (*): $p < 0.0001$, $p = 0.0005$ and $p = 0.0141$, respectively, as determined by Kruskal-Wallis test (one-way ANOVA) using Dunn's test for multiple comparison correction. Source data are provided as a Source Data file.

## Loss of RNase H1 leads to chromosome size-dependent mutagenesis

To ask if genome instability arising during short- and long-term growth after RNase H1 loss is limited to CNV and karyotype changes, we tested for the appearance of single nucleotide polymorphisms (SNPs) and small insertions and deletions (InDels). For this, we identified new SNPs and InDels that arose during growth between passage four and passage seven for induced and uninduced *RNase H1$^{Flox}$* cells, as well as in *KO* cells relative to uninduced *RNase H1$^{Flox}$* cells at passage four.

First, we examined the density of these mutations at inter-CDS regions, where R-loops are enriched (Fig. 1C) and observed the mutation patterns differed when comparing short- and long-term growth in the absence of RNase H1: a more pronounced accumulation of SNPs in these regions was seen in the induced *RNase H1$^{Flox}$* cells (Fig. 6A), while an increased accumulation of InDels was seen in the *KO* cells (Fig. 6D). In both cases, SNPs and InDels accumulated asymmetrically around

many CDSs, which we speculate may be due to differing orientations of collisions between the transcription and DNA replication machineries, leading to uneven accumulation of R-loops[134]. We also observed a slight increase in transitions relative to transversions in the *KO* cells compared to the uninduced and induced *RNase H1$^{Flox}$* cells (Supplementary Fig. 15A). pLogo enrichment analysis[135] confirmed this by showing enrichment of mutations in T and A residues, and further revealing that these mutations were not randomly distributed, but were preferentially flanked by G or C residues in the *KO* cells (Supplementary Fig. 15B).

To understand how the SNPs and Indels arise, we compared SNP and InDel levels with locations of DRIP-seq peaks (Fig. 6B, E), which revealed that while a significant fraction of R-loops are pronounced locations of these mutations, the majority are not. To understand this dichotomy, we next compared DRIP-seq and RNase H1-HA ChIP-seq signals at regions with and without SNP and InDel accumulation

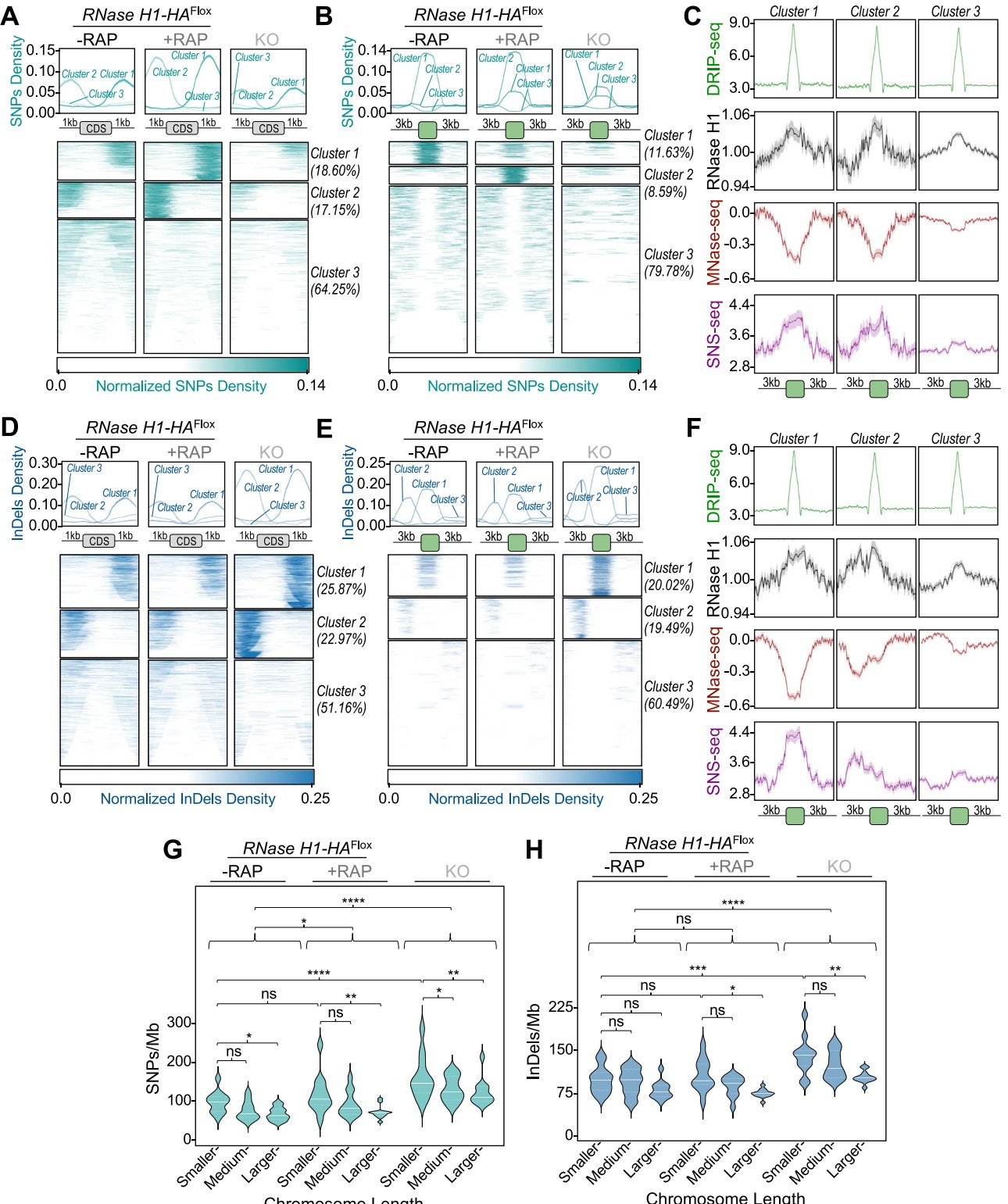

**Fig. 6 | Analysis of SNPs and InDels events upon DiCre-mediated *RNase H1* gene deletion. A, D** Metaplots (top) and colourmaps (bottom) showing normalized density of new SNPs and InDels, respectively, around annotated coding sequences (CDSs). **B, E** Metaplots (top) and colourmaps (bottom) showing normalized density of new SNPs and InDels, respectively, around DRIP-seq peaks; regions were grouped using *k-means* clustering; percentages indicate the proportion of peaks in each cluster. **C, F** Metaplots of DRIP-seq, RNase H1-HA ChIP-seq, MNase-seq and SNS-seq signals around DRIP-seq peaks grouped in clusters 1, 2 and 3 from B and E, respectively. In metaplots from (**A** to **F**), lines indicate mean and shaded areas represent SEM. **G, H** SNP and InDel densities in chromosomes grouped by length; smaller: 0.268−0.622 Mb, medium: 0.629−0.840 Mb, larger: 0.913−2.68 Mb; (****), (***), (**), (*) and ns: $p < 0.0001$, $p = 0.0001$, $p = 0.0012$, $p = 0.0155$ and not significant, respectively, as determined by one-way ANOVA and Fisher's LSD test. Source data are provided as a Source Data file.

(Fig. 6C, F, respectively). This analysis revealed that R-loops that resulted in increased mutation levels showed greater recruitment of RNase H1 than those without mutation, a difference that was associated with increased chromatin accessibility (MNase-seq) and predicted DNA replication initiation (SNS-seq). We conclude that RNase H1 acting on R-loops is required for genome stability, and the distinct mutagenic profiles upon short- and long-term cultivation is consistent with reversion of the short-term growth defect after RNase H1 loss.

We have previously observed that the density of new SNPs that arise during *L. major* growth correlates with chromosome length[136], and therefore possibly with differential DNA replication timing. To ask how the loss of size-dependent chromosome DNA replication timing seen in the absence of RNase H1 (Fig. 4K) would affect SNPs and InDels accumulation, we examined the density of these mutations, grouping chromosomes by length (Fig. 6G, H). This analysis confirmed the chromosome length-related accumulation of these mutations in uninduced cells and revealed that this effect is more pronounced in both short- and long-term growth in the absence of RNase H1. Altogether, these analyses, when considered alongside the CNV data, illustrate that R-loops acted upon by RNase H1 have widespread effects on *L. major* genome instability, which vary in pattern across the genome and during growth in the absence of the RNase H1, reflecting the complexity of global and localized R-loop association with sequence features and chromosome size. Thus, DNA replication timing is intimately linked with *Leishmania* genome plasticity through R-loops.

## Discussion

Here, we have used DRIP-seq and genetic analysis to map the localization and function of R-loops and RNase H1 in the nuclear genome of the eukaryotic parasite *L. major*. R-loops are remarkably abundant in the parasite's genome and display a distribution that, both locally and globally, correlates with features such as chromatin accessibility, G4 formation and sequence composition, all of which show a remarkable parallel with *Leishmania*'s unconventional chromosome length-related DNA replication timing programme. Our work indicates that these correlations stem from a functional relationship between R-loops and DNA replication, since we show that loss of RNase H1 abrogates the differences in DNA replication timing between large and small chromosomes and results in genome-wide, chromosome size-dependent increases in genome instability. Since R-loops are ubiquitous epigenetic features of all genomes, we suggest that RNA-DNA hybrids may be widespread, hitherto unappreciated determinants of DNA replication programming and resulting patterns of genome variation in many eukaryotes.

The local patterns of R-loop and RNase H1 enrichment in *L. major* reveal considerable intersection with transcription. R-loop enrichment at inter-CDS regions in the *L. major* genome is remarkably similar to R-loops mapped in *T. brucei*[97,98]. Hence, this work substantiates a novel association between R-loops and pre-mRNA processing across kinetoplastids, which appears distinct from roles in transcription initiation and termination seen in other eukaryotes and reflects the ubiquity of multigenic RNA Pol II transcription in kinetoplastids[105,137]. Nonetheless, it remains unclear if kinetoplastid R-loops in inter-CDSs locations simply form as a by-product of pre-mRNA processing, perhaps due to RNA Pol slowing, or if they actively participate to concentrate or organize RNA processing factors and maximize the efficiency of co-ordinated mRNA maturation during multigenic transcription. It is conceivable that G4s, together with R-loops, are a hitherto unrecognized part of mRNA processing orchestration or act to modulate RNA Pol movement. To date, no work has localized any RNase H enzyme in a kinetoplastid genome. Here, we describe mapping of RNase H1 by ChIP-seq and show that its enrichment within the PTUs correlates with DRIP-seq at many inter-CDS loci, indicating the enzyme can act on R-loops that arise during transcription elongation, albeit in a role that appears to be influenced by chromatin levels. This correlation may

suggest these R-loops are the equivalent of class II RNA-DNA hybrids that form during transcription and have been described in other eukaryotes[138], though as noted above, they may be unique to kinetoplastids due to the ubiquitous need for pre-mRNA processing by trans-splicing and polyadenylation. More pronounced RNase H1 enrichment is seen at transcription start and stop sites in *L. major*, with the former localization consistent with the formation of class I R-loops during RNA Pol II pausing as transcription initiates[138]. These data are also consistent with R-loop[97] and RNA Pol II enrichment[139,140] at transcription start sites in *T. brucei*, indicating widely conserved RNA Pol II pausing during transcription initiation. Loss of RNase H2A in *T. brucei* leads to pronounced damage accumulation at transcription start sites[98], which is not seen in RNase H1 mutants[100]. As we have not mapped DNA damage in *L. major* RNase H1 mutants, and no analysis of *Leishmania* RNase H2 function has been described, it is too early to say if the two kinetoplast parasites differ in how RNA Pol II transcription initiates. Localization of RNase H1 at polycistronic transcription termination sites is less easy to explain, since DRIP-seq does not suggest these SSRs are pronounced sites of similarly localized R-loop enrichment in *L. major* (this work) or *T. brucei*[97].

On a global level, the unanticipated chromosome size-dependent distribution of R-loops in *L. major* has not to our knowledge been described to date in any other eukaryote, suggesting it may be a novel feature of genome biology in *Leishmania*. Moreover, this global distribution of R-loops is reflected in several other chromosome-size related genome features (Fig. 2): chromatin compaction (based on MNase-seq), G4 density (G4s-seq), putative origin density (SNS-seq), and GC content are all greater on the smaller chromosomes than the larger, while SIDER1 is more concentrated in larger chromosomes. These observations suggest that global R-loop distribution reflects activities that have resulted in a chromosome size-dependent patterning of many aspects of genome content and activity in *L. major*. One explanation that might be considered as the basis for the connection between these aspects of the genome is reduced nucleosome density in the larger chromosomes, as this may allow greater levels of R-loops to accumulate, meaning the global correlation between chromatin status and R-loops reflects the localized coordination of these features at inter-CDS regions. Such decreased chromatin compaction may also allow for better resolution of G4s structures in the larger chromosomes, leading to reduction in the replication initiation activity detected by SNS-seq (which is associated with the presence of G4s)[83]. However, what aspect of *Leishmania* biology might necessitate a gradient of nucleosome density across its chromosomes is unclear. For instance, no work has described greater levels of gene expression as chromosome size increases.

A more compelling activity to explain all the above observations is *Leishmania* DNA replication programming and, as a result, timing. We have previously documented, through MFA-seq, that coordinated initiation of DNA replication in early S phase is localized to a single locus in each chromosome of *L. major* promastigotes[81,82]. Whether each locus is just a single origin is unclear[141], as is whether or not they are the sole site of DNA replication initiation in each chromosome[83,142]. Nonetheless, programming of DNA replication to predominantly initiate from a single origin or locus per chromosome would explain the chromosome size-dependent timing of *L. major* DNA replication we describe here and previously[82]. As we have argued[70,81,143,144], it is unlikely that whole genome duplication in *Leishmania* can be accomplished during S-phase using a single origin per chromosome, and so DNA replication may be supported by further, less efficient initiation activities. Such organization could then explain *Leishmania* DNA replication timing, feeding into the other genome features we describe.

An unknown feature of the above model is the nature of any putative DNA replication initiation events beyond the predicted single MFA-seq-detectable centromeric[145] origin on each *Leishmania*

chromosome. Any explanation for this putative 'supplementary' replication activity must account for chromosome size-dependent replication timing. In this light, two possibilities may be considered. First, the MNase-seq gradient we describe may not reflect decreasing chromatin compaction as chromosome size increases, but instead that the larger chromosomes are closer to the nuclear periphery and therefore more susceptible to MNase digestion, given that the dataset was generated using isolated and permeabilised nuclei[83]. If so, chromosome positioning within the *L. major* nucleus may be a determinant of replication timing in a similar way to other eukaryotes, where less efficient, late replicating loci are at the lamina-rich nuclear periphery[28,29]. If correct, chromosome subnuclear positioning does not influence replication activation at the single, main locus in each chromosome, most likely because it corresponds to the centromere[145] and these early-acting origins may overcome spatial controls[1,9]. Instead, the use of 'supplementary' origins may be more influenced by nuclear position, explaining why SNS-seq predicts greater numbers of initiation events in the smaller chromosomes (Fig. 2). R-loops can be an impediment to DNA replication that is exacerbated upon RNase H1 loss[146]. Hence, the greater abundance of R-loops as *L. major* chromosome size increases may reflect less efficient DNA replication as initiation events detected by SNS-seq become sparser. A problem with this explanation is our demonstration that loss of *L. major* RNase H1 leads to earlier replication of the larger chromosomes, since the likely increase in R-loops would be predicted to further impede DNA replication. Hence, a second explanation for DNA replication timing in *L. major* could be that R-loops mediate DNA replication initiation, as has been suggested elsewhere[147–150]. If correct, the chromosome-size dependent enrichment of R-loops may be explained: greater numbers of R-loops are needed to support replication of the larger chromosomes, where replicative DNA synthesis from a single constitutive origin duplicates less of the molecule during S-phase compared with a smaller chromosome. In addition, this explanation would explain MFA-seq mapping after RNase H1 loss: increased levels of R-loops would allow earlier replication of the larger chromosomes, resulting in loss of size-dependent DNA replication timing. In yeast[147] and bacteria[151] priming of DNA replication upon aberrant accumulation of R-loops has been seen previously. In this regard, our analysis did not reveal global R-loop accumulation upon RNase H1 loss under normal growth conditions, but only under DNA replication stress (Fig. 4G, F). In addition, under normal growth conditions we see little overlap between RNase H1-HA and DNA replication detected by EdU (Fig. 3A-C). Thus, these data suggest two scenarios (that are not mutually exclusive). First, R-loops normally make little contribution to *Leishmania* DNA replication, but their accumulation after RNase H1 loss, leading to deprogramming of replication timing, may be due to replication activation in new regions of the genome, which could be at localized loci prone to DNA replication stress or may be widespread, given the localization of R-loops at inter-CDS regions throughout PTUs. Second, a hitherto undetected R-loop resolution-independent function of *L. major* RNase H1 may mediate the temporal order of DNA replication among chromosomes.

Further work is needed to define where and how DNA replication initiation occurs across the *Leishmania* genome, including asking why changes in chromosome replication timing upon loss of RNase H1 are not associated with detectable change in cell cycle progression. Nonetheless, our demonstration that loss of RNase H1 results in increased *L. major* genome variability in patterns reflecting local and global accumulation of R-loops, as well as chromosome ploidy change, provides a mechanistic link between RNA-DNA hybrids, differential chromosome replication timing and homeostasis of the plasticity of the parasite's genome. The accumulation of CNVs and of SNPs and InDels upon RNase H1 loss is locally most prominent in inter-CDS regions where R-loops are bound by RNase H1. Thus, these mutations may be related to activation of R-loop-mediated DNA replication, or

might reflect repair DNA synthesis by error prone[152] or repair-associated DNA polymerases[153] at sites of stress. In this regard, in *Saccharomyces cerevisiae*, loss of both RNase H enzymes leads to unscheduled DNA synthesis, which is associated with increased nuclear DNA damage[154]. R-loops can also lead to genome instability by impeding DNA replication[147] and through replication-transcription clashes[134,155]. Such processes may explain the global change in mutation patterns that are most prominent after upon RNase H1 loss, where such R-loops are less efficiently resolved. Alternatively, or in addition, earlier traversal of DNA replication across R-loops in the smaller chromosomes relative to the larger could explain the increasing burden of SNPs and InDels as chromosome size decreases. It will be interesting to determine how RNase H1 mutation affects the ability of *Leishmania* to adapt to changing environments, which has been associated with genome instability[132,156,157].

Though this study details a factor that directs *Leishmania* DNA replication programming and links it to genome stability through its action on R-loops, the biological rationale for the parasite organizing DNA replication timing based on chromosome size is unclear. In this regard, we cannot currently say in *Leishmania* if R-loops might be allele-specific and provide a form of potentially non-coding RNA-associated ASARs that contribute to replication timing[55]. However, it is notable that *L. major* chromosome 31, which is always greater than diploid[158], does not follow the global pattern of R-loop or SNS-seq density, or of chromatin accessibility. In *T. brucei*, monoallelic transcription of one the -15 telomeric VSG expression sites[159,160] is intimately tied with DNA replication timing[72], and mutation of RNase H1 or RNaseH2A impairs this expression control, an effect that is associated with increased R-loops across all VSG expression sites[98,100]. Such connections between replication asynchrony, monoallelic expression and chromatin accessibility appear to mirror activities described in mammals[161] and, thus, R-loops may have wider and so far unexplored impacts on how gene expression and DNA replication intersect in kinetoplastids.

## Methods

### Parasite culture and generation of an *RNase H1-HA^Flox^* cell line

Promastigotes derived from *Leishmania major* strain LT252 (MHOM/IR/1983/IR) were cultured at 26 °C in HOMEM medium supplemented with 10% heat-inactivated foetal bovine serum. For transfections, exponentially growing cells were electroporated using Amaxa Nucleofactor™ II (pre-set program X-001). The *RNase H1-HA^Flox^* cell line was generated in three sequential transfection and selection rounds. First, a cell line expressing DiCre from the rRNA encoding locus was established. For this, a wild type strain was transfected with pGL2339 plasmid[162], previously digested with *Pac*I and *Pme*I. DiCre-expressing clones were selected with 10 µg mL⁻¹ blasticidin. Second, this cell line was further modified to concomitantly express Cas9 and T7 RNA Pol from the β-tubulin array. For this, the DiCre-expressing cell line was transfected with plasmid pTB007[163], previously digested with *Pac*I. DiCre, Cas9 and T7-expressing cells were selected in with 10 µg mL⁻¹ blasticidin and 20 µg mL⁻¹ hygromycin. Finally, by taking advantage of the Cas9/T7 system, as previously described[163], we fused all copies of *RNase H1* with six copies of *HA* while also flanking *RNase H1-6xHA* with *LoxP* sites, in a single round of transfection. For this, ORF LmjF.06.0290 encoding *RNase H1* was PCR-amplified and cloned between the *Nde*I and *Spe*I restriction sites in the vector pGL2314[164]. This construct was used as a template in a PCR reaction to generate the donor fragment containing homology flanking arms. Following ethanol precipitation, the donor fragment was transfected together with sgRNA templates into the DiCre, Cas9 and T7-expressing cell line. *RNase H1-HA^Flox^* cells were selected with 10 µg mL⁻¹ blasticidin, 20 µg mL⁻¹ hygromycin and 10 µg mL⁻¹ puromycin. Generation of sgRNAs templates and homology arms was performed as previously described[163]. In each transfection, selection was carried out by limiting

dilution in 96-well plates in the presence of the appropriate antibiotics. Integration into the expected locus was confirmed by PCR analysis. Induction of DiCre for *RNase H1* KO was performed with rapamycin, as previously reported[164,165].

## Antibodies

Mouse anti-HA (1: 5000, Sigma), mouse anti-EF1α (1: 40 000, Merck Millipore), anti-BrdU clone B44 (1: 500, BD Bioscience), and anti-DNA-RNA hybrid clone S9.6 (1:500, Sigma) primary antibodies were used here. Goat anti-Mouse IgG HRP-conjugated (ThermoFisher), goat anti-Mouse IgG Alexa Fluor 488-conjugated (ThermoFisher) and goat anti-Mouse IgG Alexa Fluor 594-conjugated (ThermoFisher) secondary antibodies were also used.

## Western blotting

Whole-cell extracts were prepared by collecting cells by centrifugation, washing with 1xPBS, resuspending in NuPAGE™ LDS Sample Buffer (ThermoFisher) supplemented with 5% β-mercaptoethanol, and heating to 95 °C for 10 min. Whole-cell extracts were resolved on 4–12% gradient Bis-Tris Protein Gels (ThermoFisher) and transferred to Polyvinylidene difluoride (PVDF) membranes (GE Life Sciences). Membranes were first blocked with 10% non-fat dry milk dissolved in 1xPBS supplemented with 0.05% Tween-20 (PBS-T) for 1 h at room temperature. Next, membranes were probed with primary antibody for 2 h at room temperature, diluted in PBS-T supplemented with 5% non-fat dry milk. After extensive washing with PBS-T, membranes were incubated with HRP-conjugated secondary antibodies in the same conditions as the primary antibodies. Finally, membranes were extensively washed with PBS-T. To detect and visualize bands, membranes were incubated with ECL Prime Western Blotting Detection Reagent (GE Life Sciences) and exposed to Hyperfilm ECL (GE Life Sciences).

## Detection of cells in S phase using flow cytometry

Exponentially growing cells were incubated with 150 µM BrdU for 30 min, collected by centrifugation and washed with 1xPBS. Washed cells were resuspended in ethanol:1xPBS (7:3) and then incubated at −20 °C for at least 16 h. Fixed cells were collected by centrifugation, washed with washing buffer (1xPBS supplemented with 1% BSA) and subjected to DNA denaturation with 2 N HCL for 30 min at room temperature. The reaction was neutralized with phosphate buffer (0.2 M Na$_2$HPO$_4$, 0.2 M KH$_2$PO$_4$, pH 7.4), cells were collected by centrifugation, and further incubated with phosphate buffer at room temperature for 30 min. Next, phosphate buffer was removed and cells were incubated with anti-BrdU antibody diluted in washing buffer supplemented with 0.2% Tween20 for 1 h at room temperature. Cells were washed with washing buffer, then incubated with anti-mouse secondary antibody conjugated with Alexa Fluor 488 diluted in washing buffer supplemented with 0.2% Tween20 for 1 h at room temperature, followed by multiple washes with washing buffer. Finally, cells were incubated for 30 min at room temperature with 10 µg mL$^{-1}$ Propidium Iodide and 10 µg mL$^{-1}$ RNase A diluted in 1xBPS, passed through a 35 µm nylon mesh and examined with FACS Celesta (BD Biosciences). Further data analysis and processing was performed with FlowJo software. Negative controls, in which anti-BrdU antibody was omitted, were used to discriminate BrdU-positive and BrdU-negative events.

## Detection of cells in S phase and RNase H1-HA using microscopy

Exponentially growing cells were incubated with 150 µM of EdU (Click-iT; Thermo Scientific) for 30 min, collected by centrifugation and washed with 1xPBS. Then, cells were fixed with 3.7% paraformaldehyde at room temperature for 15 min, collected by centrifugation and washed with 1xPBS. Fixed cells were adhered to poly-L-lysine coated slides followed by permeabilization with 0.5% TritonX-100 at room temperature for 20 min. Cells were washed with 1xPBS supplemented

with 3% BSA and subjected to Click-iT reaction, according to the manufacturer's instructions. After completion of the Click-iT reaction, cells were blocked with 1xPBS supplemented with 3% BSA for 1 h at room temperature followed by washing with 1xPBS. Cells were incubated with anti-HA antibody diluted in 1xPBS supplemented with 1% BSA and 0.01% Tween20 for 2 h at room temperature. Cells were washed with 1xPBS and further incubated with anti-mouse secondary antibody conjugated with Alexa Fluor 594 diluted in 1xPBS supplemented with 1% BSA and 0.01% Tween20 for 1 h at room temperature. After washing cells with 1xPBS, DNA was stained with DAPI. Images were acquired with a Zeiss Elyra Super-resolution microscope and further processed with ImageJ software.

## Detection of R-loops using microscopy

$2 \times 10^7$ exponentially growing cells were collected by centrifugation and resuspended in 1.5 ml of 1xPBS supplemented 5 mM EDTA. Next, 3.5 mL methanol was added slowly, and the cell suspension was incubated for 2 h at 4 °C under constant agitation. Cells were collected by centrifugation and washed with 1xPBS supplemented with 5 mM EDTA. Next, cells were resuspended in 1xPBS supplemented with 0.5% TritonX-100 and incubated on ice for 10 min. After centrifugation, cells were resuspended in staining buffer (1xPBS supplemented with 0.01% Tween20 and 0.1% BSA) and split into two aliquots. To one of the aliquots, 5 mM MgCl$_2$ and 2 U of recombinant *Escherichia coli* RNase HI (NEB) were added. Both aliquots were incubated at 37 °C for 1 h under constant agitation. Cells were collected by centrifugation and incubated with S9.6 antibody diluted in staining buffer at 37 °C for 1 h under constant agitation. Cells were washed with staining buffer, resuspended in 1xPBS and adhered to poly-L-lysine coated slides at room temperature for 30 min. DNA was stained with DAPI and images acquired with Zeiss Elyra Super-resolution microscope or Leica DMi8 microscope. Further image processing was performed with ImageJ software.

## DNA-RNA hybrid immunoprecipitation and sequencing (DRIP-seq)

$2 \times 10^9$ exponentially growing cells were collected and washed with 1xPBS. Cells were resuspended in lysis buffer (10 mM TrisHCl pH8.0; 100 mM NaCl; 25 mM EDTA) at the concentration of $1 \times 10^8$ cells mL$^{-1}$. The cell suspension was supplement with 0.1 mg mL$^{-1}$ proteinase K and 0.5% SDS and incubated overnight at 37 °C. Nucleic acids were extracted with equal volume of ultrapure phenol:chloroform:isoamylic alcohol (25:24:1) (ThermoFisher) followed by extraction with chloroform alone. Then, nucleic acids were precipitated by adding 0.1 V of 3 M sodium acetate pH5.2 and 2 V of ice-cold absolute ethanol, followed by centrifugation at 16,000 *g* under refrigeration for 10 min. The pellet was resuspended in 1.5 mL ice-cold 75% ethanol and centrifuged for 10 min under refrigeration. Next, the pellet was dissolved in 0.3 mL of ultrapure TE pH 8.0 (ThermoFisher) and incubated at 4 °C for 12 h. The final nucleic acid suspension was carefully homogenized and divided into two aliquots. One aliquot was left untreated, and the other was treated with 20 U of recombinant *Escherichia coli* RNase HI (Invitrogen) overnight at 37 °C. Nucleic acids were extracted once with an equal volume of ultrapure phenol:chloroform:isoamylic alcohol (25:24:1) and precipitated with 0.1 V 3 M sodium acetate pH5.2 and 2 V of ice-cold 100% ethanol. Pellets were resuspended and subjected to digestion at 37 °C for 16 h with one of the following digestion cocktails: Cocktail 1 (*Hind*III, *Eco*RI, *Bsr*GI, *Xba*I and *Ssp*I) was used for DRIP-seq data shown in main figures, and Cocktail 2 (*Hind*III, *Eco*RI, *Bsr*GI, *Xba*I and *Ssp*I *Bam*H1, *Nco*I, *Apa*LI and *Pvu*II) was used for DRIP-seq data shown in Supplementary Figs. The digestion cocktail for RNase HI-negative samples was further supplemented with 120 U RNaseOUT (Invitrogen), while the digestion cocktail for RNase HI-positive samples were supplemented with 10 U recombinant *Escherichia coli* RNase HI (Invitrogen). After digestion, nucleic acids were extracted with an

equal volume of ultrapure phenol:chloroform:isoamylic alcohol (25:24:1), precipitated with 0.1 V 3 M sodium acetate pH5.2 and 2 V of ice-cold 100% ethanol. Pellets were resuspended in 0.1 mL of ultrapure TE pH 8.0 and 5% of the final solution was saved as input. DNA concentration was determined using Qubit (Invitrogen) and 40 ug DNA was diluted in 0.9 mL of IP buffer (10 mM Sodium Phosphate pH 6.8; 140 mM NaCl; 0.5% Triton X-100). This solution was mixed with 0.4 mL of Dynabeads M-280 Sheep Anti-Mouse IgG (Invitrogen) previously conjugated with 25 ug of S9.6 antibody (Millipore) and incubated overnight at 4 °C under constant agitation. Beads were collected using a magnetic rack and washed four times with IP buffer for 10 min each at 4 °C under constant agitation. Then, beads were resuspended in 0.2 mL of 1xPBS supplemented with Proteinase K from the DNeasy Blood and Tissue Kit (QIAGEN) and incubated at 37 °C in a thermomixer for 4 h. Samples were centrifuged and supernatant was collected, then DNA was extracted following DNeasy Blood and Tissue Kit standard protocols. Library preparation and sequencing is described below.

Enrichment of immunoprecipitated material over input is expressed as ratios and was determined using bamCompare (DeepTools) over a 60 bp rolling window. For this, only reads with mapping quality >10 were considered. Libraries sizes were normalized using the SES method, pair-ended extension was employed, PCR duplicates were ignored and regions were centred with respect to the fragment length. All snapshot representations were performed using Gviz[166]. In the DRIP-seq analysis, peaks were identified based on regions exhibiting an enrichment greater than 2.5-fold in the DRIP material compared to the input control, as well as a more than 2.5-fold enrichment in samples untreated with RNase H compared to those treated with RNase H. Regions smaller than 60 bp were excluded from the analysis, and adjacent regions within 60 bp of each other were merged.

### Chromatin immunoprecipitation (ChIP-seq)
Exponentially growing RNase H1-HA^Flox cells were collected by centrifugation, resuspended in 1xPBS supplemented with 1% paraformaldehyde and incubated at room temperature for 15 min under constant agitation. Cells were collected and resuspended in lysis buffer (100 mM Tris pH 8.8; 200 mM NaCl; 1% NP40; 10% glycerol; 10 mM EDTA; 5 mM PMSF; 5 mM 1,10-phenantroline; 2x EDTA-free Pierce™ Protease Inhibitor) and subjected to 10 rounds of sonication of 20 sec each. Lysates were clarified by centrifugation and 5% was saved as input. The remainder of the lysate was incubated with 0.1 mL of Dynabeads M-280 Sheep Anti-Mouse IgG (Invitrogen) previously conjugated with 10 ug of anti-HA antibody (Millipore) and incubated at 4 °C for 16 h under constant agitation. Beads were collected using a magnetic rack and washed three times with lysis buffer. Next, beads were resuspended in elution buffer (50 mM Tris pH 7.6; 1% SDS) and incubated at 55 °C for 10 min under constant agitation. Crosslink reversal of eluted and input materials was performed by incubating samples at 65 °C for 12 h. DNA was further purified using DNeasy Blood and Tissue Kit (QIAGEN) following the manufacturer's instructions. Library preparation and sequencing is described below.

### Replication timing profiling using Marker Frequency Analysis coupled with Illumina sequencing (MFA-seq)
Genomic DNA was extracted from exponentially growing and stationary cells using DNeasy Blood & Tissue Kit (QIAGEN). After library preparation and Illumina sequencing (see details below) BamCompare (DeepTools) was used to determine reads abundance from exponentially growing cells relative to the reads from stationary culture. Ratios was first calculated in 1 kb consecutive windows using the reads counts method for normalization. Raw ratios were further transformed into Z scores relative to the whole genome ratios average as calculated in 15 kb sliding windows. MFAseq snapshots were represented in a graphical form using Gviz[166].

### Library preparation, sequencing and analysis
All libraries were prepared using using QIAseq FX DNA Library Kit (QIAGEN) and were sequenced as 75 nucleotide paired-end reads. Sequencing was performed at Glasgow Polyomics (www.polyomics.gla.ac.uk/index.html) using a NextSeq™ 500 Illumina platform. The Galaxy web platform (usegalaxy.org)[167] was used for most of the downstream data processing. For quality control and removal of adaptors, FastQC (http://www.bioinformatics.babraham.ac.uk/projects/fastqc/) and trimomatic[168] were used, respectively. Trimmed reads were mapped to the reference genome (*Leishmania major Friedlin* v39, available at Tritrypdb - http://tritrypdb.org/tritrypdb/) using BWA-mem[169].

### Cell cycle synchronization and DNA content analysis using fluorescent activated cell sorting (FACS)
Exponentially growing cells were incubated with 5 mM hydroxyurea for 8 h. Then, cells were washed and re-seeded into hydroxyurea-free medium and collected by centrifugation every 3 h. Cells were washed in 1×PBS supplemented with 5 mM EDTA, collected by centrifugation, resuspended in methanol:1xPBS (7:3) and then incubated at 4 °C for at least 2 h. Fixed cells were collected by centrifugation, washed with 1×PBS supplemented with 5 mM EDTA and incubated for 30 min at room temperature with $10 \, \mu g \, mL^{-1}$ Propidium Iodide and $10 \, \mu g \, mL^{-1}$ RNase A diluted in 1×PBS containing 5 mM EDTA. Cells were then passed through a 35 μm nylon mesh and examined with FACS Celesta (BD Biosciences). Further data analysis and data processing was performed with FlowJo software.

### SNPs, InDels and CNV analysis
SNPs and InDels relative to the reference genome were detected in P4, P7 and in KO cells using freeBayes[170]. Only those SNPs and InDels in regions with read depth of at least 5, with at least 2 supporting reads, and a map quality of 30 were considered. To better capture the genomic variability in the time frame of the experiments, variants present simultaneously in P4 and P7 or P4 and KO cells were excluded from the analysis using VCF-VFC intersect function from VCFtools package[171]. The SNP density function from VCFtools was used to calculate SNPs and InDels density in consecutive 1 Kb windows. Genome-wide CNVs were calculated with bamCompare (DeepTools), using the RPKM normalization method. Pair-ended extension was employed, PCR duplicates were ignored, and regions were centered with respect to the fragment length. Fold-change was expressed as log2.

### Reporting summary
Further information on research design is available in the Nature Portfolio Reporting Summary linked to this article.

## Data availability
Sequences used in this study have been deposited in the EMBL-EBI European Nucleotide Archive (ENA) under the accession number PRJEB75366; MFA-seq data is available in the NCBI Sequence Read Archive (SRA) with the accession number PRJNA1108605. Source data are provided with this paper.

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

## Acknowledgements

The authors thank all current and previous members of the McCulloch lab for input. This work was supported by the Wellcome Trust [224501/Z/21/Z to R.M., 218648/Z/19/Z to E.B.], the BBSRC [BB/N016165/1, BB/R017166/1 to R.M., BB/W001101/1 to R.M. and C.M.], the MRC [MR/S019472/1 to R.M. and J.D.], and the European Union's Horizon 2020 research and innovation programme under the Marie Sklodowska-Curie grant agreement No 750259 [Individual Fellowship, RECREPEMLE to

J.D.]. The Wellcome Centre for Integrative Parasitology was supported by core funding from the Wellcome Trust [104111 to R.M.].

## Author contributions

Conceived the study: J.D., E.B., R.M. Designed and conducted the research: J.D., M.K., C.M., C.L. Analyzed the data: J.D., E.B., M.K., C.M., R.M. Wrote the initial draft: J.D., R.M. Edited and approved paper: J.D., E.B., M.K., C.M., R.M. Funding: J.D., E.B., C.M., R.M.

## Competing interests

The authors declare no competing interests.
