## [Transparent Peer Review file · Nature Communications]

R-loops acted on by RNase H1 influence DNA replication timing and genome stability in Leishmania

Corresponding Author: Professor Richard McCulloch

Version 0:

Reviewer comments:

Reviewer #1

(Remarks to the Author)
Comments for the Author:

In this study, the authors first profiled the landscape of R-loops in the nuclear genome of the eukaryotic parasite *Leishmania major*, and revealed that the R-loop distribution correlates with genome features such as chromatin accessibility, G4 formation and sequence composition. Strikingly, they found that R-loops accumulate in chromosomes in a pattern that reflects their replication timing, and R-loops are more enriched in chromosomes with large size. Further, by conditional knockout of RNase H1, they showed that loss of *L. major* RNase H1 results in a transient growth defect, and the R-loops significantly accumulate under DNA replication stress resulting from hydroxyurea (HU) exposure upon both short- and long-term RNase H1 loss. By investigating the effect of RNase H1 loss through MFA-seq, they found that RNase H1 activity may be essential to replication timing maintenance in *L. major*. Finally, they revealed that loss of RNase H1 leads to chromosome size-dependent mutagenesis and genome instability.

Collectively, this work reveals that R-loops and RNase H may be critical to DNA replication timing and genome instability in *Leishmania*. These findings will contribute to understanding how the chromosome size dependent replication timing might arise, and should of great interest for the readership of Nature Communications.

Weaknesses

Lines 308-309: Please be careful with your conclusions here, especially the apparent loss of CNV found in regions of R-loop accumulation in RNase H1-KO lines (Lines 352-355).

Lines 141-152: These paragraphs should be in the introduction section.

Line 197: Please give full name of SNS-seq in parentheses at first use.

Line 214: "Thus, the presence of G4s is a good predictor of R-loop accumulation at the local level, but not at a genome-wide level."

Please explain why G4s is not correlated with R-loops here, and I can't find that the presence of G4s is a good predictor of R-loop accumulation at the local level in the heatmaps.

Lines 246-251: This section should be placed after Line 262, and integrated with Lines 253-272.

Lines 297-300: Fig. 3A showed that RNase H1 was primarily enriched in non-replicating cells, so which cell phase did the authors detect in the experiment of Fig. 4F?

Lines 381-384: Please explain this phenomenon. Is this because SNPs accumulated in the short-term growth have further mutated into InDels in the long-term growth?

Line 750: Chromosomes are ordered by size in ascending order?

Line 754: distribution patterns of DNA replication timing determined by MFA-seq?

Line 754: “chromatin accessibility determined by MNase-seq, G4s and short interspersed degenerate retroposons 1 (SIDER1), respectively;” should be “chromatin accessibility determined by MNase-seq, G-Quadruplex density determined by G4-seq, and distribution of direct and inverted repeat DNA sequences determined by short interspersed degenerate retroposons 1 (SIDER1), respectively;”.

Figure 2A: Please explain in figure legend what “Shuffled” stands for.

The exception of chromosome 31, which does not follow the pattern of all other chromosomes, should be mentioned in the results.

Figure 2D: Please note the title “MNase-seq”, and should explain the number like Fig.2B, for example “compacted-relaxed”.

Figure 5E, F: Heatmaps should be better labeled.

Reviewer #2

(Remarks to the Author)

The manuscript concerns R-loops in the Leishmania genome and presents various sequence analyses to argue that R-loops are an additional part of genome patterning that is integral to the programming of DNA replication. The authors also argue that R-loops provide a mechanistic link to adaptive genome plasticity in Leishmania. The topic is important fundamental biology, likely crucial to Leishmania genome and cell function, and not much is known. The study is very well considered and generally meticulous in its approach. It uses multiple, orthogonal sequencing techniques to characterise genome patterning in new ways and, in the experimental high point, it uses CRISPR/Cas9 to flank the endogenous RNase H1 ORF with loxP sites, allowing the gene to be deleted by induction of DiCre activity.

While the title emphasises the deterministic action of R-loops (as it turns out, among many other things) in effecting the timing of replication and genome stability, the experimental evidence relates strictly to RNase H1. They conclude that RNase H1 provides nuclear functions in Leishmania but does not appear to be linked to replicative DNA synthesis, but that when cell cycle progression at G1/S is blocked in RNase H1 null mutants, R-loops accumulated. I found it difficult to reconcile and understand exactly what role in DNA synthesis the authors consider RNase H1 to have. I invite them to point to the relevant statement in the manuscript.

What comes across strongly (and perhaps laboriously) are the many coincidences between R-loops and other patterning, e.g. R-loops and the maturation of mRNA (l180). Besides the direct effects of RNase H1 knock-out, I am not sure that other observations go beyond description. There is a trail of correlations between different phenomena but there is no evidence for the mechanism by which R-loops initiate DNA replication, if that is the authors contention.

R-loops may be just another aspect of size-dependent genome patterning that (somehow) controls the timing of DNA replication. They seem to understand this: “These observations suggest that global R-loop distribution reflects activities that have resulted in a chromosome size-dependent patterning of many aspects of genome content and activity in *L. major*” but elsewhere seem to emphasise the crucial role of R-loops “DNA replication timing is intimately linked with Leishmania genome plasticity through R-loops”. I invite the authors to comment on whether they think R-loops are an especially significant part of genome patterning. Are they a cause of anything, or simply an effect of the ‘activities’ that produce genome patterning? I invite them to point to relevant statement in the manuscript.

Minor points:

- The introduction needs to explain what an R-loop is and how it relates to DNA/RNA hybrid structures? Why should a cell need to remove them?
- The introduction might also briefly explain what the bigger picture is here. The manuscript drives at the idea that the timing of replication within genomes is flexible and adaptive. Better say that explicitly.
- l106, typo, schizogony
- l363: typo, aneuploid
- l1394-5: “We have previously observed that SNP occurrence among chromosomes correlates with their length” – what is meant by ‘occurrence’ and what form of correlation is observed?
- Throughout, be more precise in describing what is being measured. DRIP-seq ‘pattern’ (l206), R-loop ‘accumulation’ (l215), DRIPseq ‘levels’ (l226), ‘accumulation’ (l268). Is it most precise to say ‘R-loop density’? Use a consistent term that makes it clear that the number of R-loops is corrected for chromosome size.

Reviewer #3

(Remarks to the Author)

In this manuscript, Damasceno et al., profile both R-loop and RNase H1 genomic distributions in the *Leishmania major* genome and show that these accumulate in a chromosome length-dependent manner that itself is negatively correlated with replication timing. Using conditional gene KO, they further show that RNase H1 loss leads to transient growth perturbation and a flattening of the replication timing differences between short and long chromosomes. In addition, RNase H1 KO leads to increased frequencies of CNVs, Indels and SNPs.

Despite reporting some interesting results on an interesting topic, the manuscript suffers from a number of significant flaws, including a tendency to over-interpret findings. The overall message that is encapsulated in the title, namely that “R-loops acted on by RNase H1 are a determinant of chromosome length-associated DNA replication timing and genome stability in *Leishmania*” is unsupported by evidence. The claim R-loops are acted upon by RNase H1 is unsupported because authors do not present any overlap analysis between their R-loop maps and RNase H1 maps. We are told that they RNase H1 enriches at regions where R-loops are found to accumulate without any direct comparisons being performed. We are told that “R-loops acted upon by RNase H1 have widespread effects on *L. major* genome instability” without any direct high-res comparison of R-loops maps with CNV, Indels or SNPs maps. Finally, we are told that “R-loops are determinant of the replication timing” without being provided evidence of overlap between R-loops, RNase H1 and replication origins. This is all the more puzzling that the data is at hand. Overall, it is clear that RNase H1 has a strong effect on the replication program and on genome instability. I am much less convinced that R-loops have anything to do with this. If the authors want to push the message that R-loops are involved then they need to show that 1) RNase H1 binding sites overlap strongly with (at least a subset) of R-loops; and 2) these RNase H1-bound R-loops are exactly where CNVs and InDels/SNP accumulate.

Major comments:

1- The authors map R-loops in *Leishmania* using DRIP-seq. A key finding is that “R-loops display pronounced accumulation at intergenic regions between coding sequences (CDSs) in polycistronic transcription units”. R-loops in yeasts and mammalian cells clearly occur co-transcriptionally. Could the authors please provide transcription data (ideally nascent) to indicate whether intergenic regions are transcribed? If not, how do they envision R-loops forming?? The authors should add correlation with transcription on their Figure 2 set of panels.

2- Authors initially indicate that R-loops correlate with higher G4 occurrence but motif analysis (1E) doesn't reveal the presence of that motif. Why? In fact, motif analysis seems to indicate that R-loop-associated sequence features are opposite that of G4s. Figures 2A, 2E and 2R show an inverse correlation between G4s and R-loops. This needs to be clarified. The statement that “Thus, the presence of G4s is a good predictor of R-loop accumulation at the local level, but not at a genome-wide level” (line 215) is puzzling especially given the preceding statement that “Re-analysis of G4-seq data revealed a similar and significant anticorrelation with DRIP-seq”.

The identification of TCn and CCTn as R-loop-enriched motifs is very puzzling as these sequences are among the absolute worst in terms of RNA:DNA hybrid stability. R-loops in yeasts and mammalian cells have long been associated with purine-rich regions, mostly G-rich. How do the authors explain this? Have they provided evidence that a pyrimidine rich regions, when cloned in an in vitro transcription plasmid vector, can form R-loops? I doubt it. This raises significant questions. Could the issue here be one of strandedness? The maps produced here are not stranded (which is too bad). Is it possible that transcription runs antisense to that of the transcription units? This would restore purine-rich motifs as R-loop-associated instead of pyrimidine-rich motifs. Finally, the observation of an anti-correlation between DRIP signals and GC fraction is very counter-intuitive. I wonder if the observation of a positive correlation with SIDER1 retroposons provides an answer to some of the questions raised here. Namely, could it be possible that DRIP detects RNA:DNA hybrids that are the products of a retrotransposition event? Previous mapping of R-loops in yeast showed that Ty element activity generates signals that can be captured in DRIP (PMID: 25357144). This may also explain the association between DRIP signals and intergenic regions. The authors should provide additional information about this possibility.

3- One of the key claims of the MS, namely that “R-loops are acted on by RNase H1” clearly predicts that RNase H1 binding sites should overlap with R-loops. Nowhere is a detailed overlap analysis of these two datasets provided. This is unacceptable. DRIP-seq tracks should be shown for Figure 3B and 3C. Nowhere is it stated that R-loops correlate with AcH3 or base J, which seem to correlate very strongly with RNase H1 binding sites. The statement that “the global distribution of *L. major* RNase H1 mirrors that of the R-loops that it acts to dissolve” might be true at a bird's eye view but the devil is in the detail and high resolution comparisons should be provided to back this key claim.

4- The observation that R-loop signals measured by S9.6 immunofluorescence increase upon HU treatment but not under any other tested conditions is interesting but raises quite a few other questions. I question the reliability of the assay. The authors purport to show that S9.6 IF works in *Leishmania* when it is known not to work in mammalian cells (Figure 1A). I wish I could believe this but for this, I would need to see additional controls such as RNase T1 insensitivity. I also believe the authors should add MgCl₂ to the mock treated sample in case Mg ions facilitates RNA degradation by endogenous ribonucleases; having one sample receiving 5 mM MgCl₂ and RNase H1 and the other sample receiving nothing is not sufficient. Given the concerns about S9.6 imaging, I believe the authors should perform DRIP-seq in H1 KO cells to substantiate their claims.

5- The observation of a flattened replication program with increased CNVs and InDels in the absence of RNase H1 is interesting. Mechanistically, however, how this arises remains obscure. If the authors want to convince that R-loops “acted upon by RNase H1” have a hand in causing genome instability, they again need to compare the CNV maps to the R-loop DRIP maps directly and at high resolution. Asking readers to visually compare Figure 5B (CNVs) to Figure 2A is totally

inappropriate. It seems to me that RNase H1 has a clear hand in regulating the replication program in Leishmania (how?) but I am quite unclear that R-loops have anything to do with it. For this correlation to be better documented, authors need to show that 1) RNase H1 binding sites overlap strongly with (at least a subset) of R-loops and 2) these RNase H1-bound R-loops are exactly where CNVs and InDels/SNP accumulate. The authors have the data at hand. That they do not provide a direct comparison of these datasets when it is so relevant to their main claims is raising questions.

Other comments:

- The manuscript is very richly referenced (169 in total!). The Introduction is very detailed – is this all necessary? The authors map R-loops in Leishmania using DRIP-seq. Ironically, a reference to the original DRIP-seq paper is not provided.
- In reference 87, this work only shows that R-loops occur at sites of rearrangements in CSR, not that they are required. The role, if any, of R-loops in CSR remains controversial. Please adjust.
- Figure 1D depicts the R-loop peak length frequency distribution. This most likely reflects the length distribution of the DNA fragments generated by restriction enzymes, not the length of R-loops. In DRIP-based assays, it is well known that the entire fragment is IP'ed and that the resolution of the method is dictated by the enzymes used for genome fragmentation. I suggest removing, or properly explaining.
- On figure 3A, the authors need to quantify the overlap between RNase H1 and EdU signals for at least 50 independent cells to conclude that RNase H1 “showed a higher abundance in non-replicating cells”. Showing two cells is not sufficient or rigorous.
- What is being shown at the bottom of Figure 4H? I thought Leishmania chromosome were replicated from a single-origin per chromosome located to subtelomeres (Introduction). Are the PTUs shown below subtelomeric? If not, how were they chosen? If origins map to SSRs, which are common, then there isn't a single origin per chromosome. Please indicate a scale! Are we looking at a whole chromosome? This is confusing. How are early and late-replicating regions defined?

Version 1:

Reviewer comments:

Reviewer #1

(Remarks to the Author)

The authors have addressed my concerns from the initial round of review.

Reviewer #2

(Remarks to the Author)

Reviewer #3

(Remarks to the Author)

I appreciate the authors' answers to my queries and believe the manuscript has been significantly improved as a result.

I am still puzzled by the sequence dependence of R-loops in inter-CDS regions. While the new data and clarified language helps, the legend to Figure S2C should be significantly improved prior to acceptance. As it is, I do not know what I am looking at (what are the %ages referring to? Why the two distinct arrows to the right and left? How do the motifs relate to the inter-CSD regions? Shouldn't the motifs appear in a reverse complement manner and order for the right and left arrows?). This is important since R-loop formation over pyrimidine-rich sequences would really be a departure for the R-loop field in terms of nucleic acid properties. If true, this may point to specific mechanisms at play to either form or stabilize these unusual R-loops.

The mutual relationships, at least in terms of localization, between RNase H1 and R-loops have been clarified. Significant questions surrounding the striking enrichment of RNase H1 at SSRs together with base J and Ach3 remain. Similarly, the observation that a majority of DRIP peaks are RNase H1 free raise questions. I suppose these could be explored in the future and feel confident that the present study now presents a more complete set of observations.

I particularly appreciate the improvements to the section on RNase H1 binding, R-loops and CNVs, InDels, SNPs. The manuscript is stronger as a result.

We thank all reviewers for their helpful comments and very useful suggestions. Detailed responses to all the reviews are provided (red text), and below is a summary of the new data and analysis provided:

Fig.1. New panel, revealing putative G4 motifs at sites of R-loop enrichment.

Fig.3. New analysis of correspondence between R-loops (DRIP-seq) and RNase H1 (RNase H1-HA CHIP-seq). New quantification of EdU and anti-HA (RNase H1-HA) signals in IF images of multiple cells.

Fig.5. New analysis of correspondence between CNV events during *L. major* growth and locations of R-loops (DRIP-seq) and RNase H1 (RNase H1-HA CHIP-seq).

New Fig.6. New analysis of correspondence between SNP and InDel events during *L. major* growth and locations of R-loops (DRIP-seq) and RNase H1 (RNase H1-HA CHIP-seq).

New Fig.S1. New IF analysis in *L. major* using the S9.6 antibody and including MgCl₂ and RNaseT1 treatments as controls.

New Fig.S2. Mapping of nascent RNA (PRO-seq) and mature mRNA (RNA-seq), demonstrating continuous transcription across *L. major* polycistronic transcription units, including in inter-CDS regions where R-loops are found.

New Fig.S7. Testing correlation between chromosome-size and nascent or mature RNA levels in *L. major*.

New Fig.S8. Testing for localisation of R-loops at known *L. major* retrotransposon elements and other repeats.

New Fig.S13. An explanation of how DNA replication timing is inferred from MFA-seq data.

Fig.S15. New analysis of overlap between new SNPs and Indels.

REVIEWER COMMENTS

Reviewer #1 (Remarks to the Author):

Comments for the Author:

In this study, the authors first profiled the landscape of R-loops in the nuclear genome of the eukaryotic parasite *Leishmania major*, and revealed that the R-loop distribution correlates with genome features such as chromatin accessibility, G4 formation and sequence composition. Strikingly, they found that R-loops accumulate in chromosomes in a pattern that reflects their replication timing, and R-loops are more enriched in chromosomes with large size. Further, by conditional knockout of RNase H1, they showed that loss of *L. major* RNase H1 results in a transient growth defect, and the R-loops significantly accumulate under DNA replication stress resulting from hydroxyurea (HU) exposure upon both short- and long-term RNase H1 loss. By investigating the effect of RNase H1 loss through MFA-seq, they found that RNase H1 activity may be essential to replication timing maintenance in *L. major*. Finally, they revealed that loss of RNase H1 leads to chromosome size-dependent mutagenesis and genome instability.

Collectively, this work reveals that R-loops and RNase H may be critical to DNA replication timing and genome instability in Leishmania. These findings will contribute to understanding how the chromosome size dependent replication timing might arise, and should of great interest for the readership of Nature Communications.

Weaknesses

Lines 308-309: Please be careful with your conclusions here, especially the apparent loss of CNV found in regions of R-loop accumulation in RNase H1-KO lines (Lines 352-355).

Thank you for this suggestion: please see the responses to reviewer #3, *where we describe new analysis (Fig.3D-I, Fig. 5C-E and new Fig. 6B,C,E,F)* that more clearly documents RNase H1 enrichment at a subset of R-loops (at SSRs and within PTUs; Fig.3), and we show that CNVs and SNP/InDels that accumulate after RNase H1 loss at locations where R-loops are bound by RNase H1.

Lines 141-152: These paragraphs should be in the introduction section.

Respectfully, we disagree; we prefer the current organisation, with a focus on DNA replication timing/programming in the introduction.

Line 197: Please give full name of SNS-seq in parentheses at first use.

Corrected, thank you.

Line 214: "Thus, the presence of G4s is a good predictor of R-loop accumulation at the local level, but not at a genome-wide level."

Please explain why G4s is not correlated with R-loops here, and I can't find that the presence of G4s is a good predictor of R-loop accumulation at the local level in the heatmaps.

We have explained this more fully by referring to the appropriate data that shows local correlation:

'...despite G4 and R-loop densities positively correlating with each other in specific, local genomic areas (Fig. 1C), they are negatively correlated genome-wide'.

Lines 246-251: This section should be placed after Line 262, and integrated with Lines 253-272.

I'm afraid we do not understand the rationale for this suggestion, since lines 246-251 describe IF analysis and later paragraphs describe ChIP-seq.

Lines 297-300: Fig. 3A showed that RNase H1 was primarily enriched in non-replicating cells, so which cell phase did the authors detect in the experiment of Fig. 4F?

Cells not treated with HU (-HU) are exponentially growing and therefore all cell cycle stages are represented, as indicated by FACS analysis (Fig. S14A). Cells treated with HU (+HU) were arrested at the G1/S boundary, as indicated by FACS (Fig.S14A) and as stated:

'However, when cell cycle progression at G1/S was blocked via hydroxyurea treatment ...'

Lines 381-384: Please explain this phenomenon. Is this because SNPs accumulated in the short-term growth have further mutated into InDels in the long-term growth?

I'm afraid we cannot answer the question with certainty, despite these data being analysed further in response to comments from reviewer #3 (Fig.5 and new Fig.6). InDels are more prevalent in the KO mutant than after short-term loss of RNase H1, while SNPs are more prevalent after DiCre excision, and the locations of both mutations correspond to R-loops bound by RNase H1. Nonetheless, Fig.15C shows that only a small proportion of SNPs and InDels overlap, perhaps arguing against conversion from the former to the latter.

Line 750: Chromosomes are ordered by size in ascending order?

Correct.

Line 754: distribution patterns of DNA replication timing determined by MFA-seq?

Thank you, we have clarified this:

'distribution patterns of DNA replication timing (MFA-seq),...'

Line 754: "chromatin accessibility determined by MNase-seq, G4s and short interspersed degenerate retroposons 1 (SIDER1), respectively;" should be "chromatin accessibility determined by MNase-seq, G-Quadruplex density determined by G4-seq, and distribution of direct and inverted repeat DNA sequences determined by short interspersed degenerate retroposons 1 (SIDER1), respectively;".

Thank you – corrected.

Figure 2A: Please explain in figure legend what "Shuffled" stands for.

This is now explained in the legend to Fig. 2A:

'Shuffled indicates DRIP-seq signal plotted after R-loop peaks were randomly distributed throughout the genome'.

The exception of chromosome 31, which does not follow the pattern of all other chromosomes, should be mentioned in the results.

This has been added to the text:

'...DRIP-seq density displayed a significant correlation with L. major chromosome length (Fig. 2A, H; independent replicate shown in Fig. S5A), with R-loops becoming more abundant as chromosome size increased, though it is notable that chromosome 31 did follow this pattern.'

Figure 2D: Please note the title "MNase-seq", and should explain the number like Fig.2B, for example "compacted-relaxed".

Corrected, thank you.

Figure 5E, F: Heatmaps should be better labeled.

Labels have been added. These are now in *new Fig. 6*.

Reviewer #2 (Remarks to the Author):

The manuscript concerns R-loops in the Leishmania genome and presents various sequence analyses to argue that R-loops are an additional part of genome patterning that is integral to the programming of DNA replication. The authors also argue that R-loops provide a mechanistic link to adaptive genome plasticity in Leishmania. The topic is important fundamental biology, likely crucial to Leishmania genome and cell function, and not much is known. The study is very well considered and generally meticulous in its approach. It uses multiple, orthogonal sequencing techniques to characterise genome patterning in new ways and, in the experimental high point, it uses CRISPR/Cas9 to flank the endogenous RNase H1 ORF with loxP sites, allowing the gene to be deleted by induction of DiCre activity.

While the title emphasises the deterministic action of R-loops (as it turns out, among many other things) in effecting the timing of replication and genome stability, the experimental evidence relates strictly to RNase H1. They conclude that RNase H1 provides nuclear functions in Leishmania but does not appear to be linked to replicative DNA synthesis, but that when cell cycle progression at G1/S is blocked in RNase H1 null mutants, R-loops accumulated. I found it difficult to reconcile and understand exactly what role in DNA synthesis the authors consider RNase H1 to have. I invite them to point to the relevant statement in the manuscript.

Based on our data, there is no evidence to support the direct involvement of RNaseH1 or R-loops in Leishmania DNA synthesis, but only with DNA replication timing. We do, however, propose likely scenarios

in the discussion, which are strengthened by new data (Fig.3, Fig.5, new Fig.6; see response to reviewer #3 for more detailed discussion) showing localisation of RNase H1 at many R-loops, and with mutation at these loci after RNase H1 loss:

'In yeast¹⁴⁷ and bacteria¹⁵¹ priming of DNA replication upon aberrant accumulation of R-loops has been seen previously. In this regard, our analysis did not reveal global R-loop accumulation upon RNase H1 loss under normal growth conditions, but only under DNA replication stress (Fig. 4G,F). In addition, under normal growth conditions we see little overlap between RNase H1-HA and DNA replication detected by EdU (Fig. 3A-C). Thus, these data suggest two scenarios (that are not mutually exclusive). First, R-loops normally make little contribution to Leishmania DNA replication, but their accumulation after RNase H1 loss, leading to deprogramming of replication timing, may be due to replication activation in new regions of the genome, which could be at localised loci prone to DNA replication stress or may be widespread, given the localisation of R-loops at inter-CDS regions throughout PTUs. Second, a hitherto undetected R-loop resolution-independent function of L. major RNase H1 may mediate the temporal order of DNA replication among chromosomes.'

What comes across strongly (and perhaps laboriously) are the many coincidences between R-loops and other patterning, e.g. R-loops and the maturation of mRNA (l180). Besides the direct effects of RNase H1 knock-out, I am not sure that other observations go beyond description. There is a trail of correlations between different phenomena but there is no evidence for the mechanism by which R-loops initiate DNA replication, if that is the authors contention.

We accept this criticism, but please see the comment above: by adding new analysis (Fig.3, Fig.5, new Fig.6) we are now able to suggest mechanistic links between localisation of RNase H1 to R-loops and changes in DNA replication timing and the emergence of mutants (CNV, SNPs and InDEls) after loss of RNase H1.

R-loops may be just another aspect of size-dependent genome patterning that (somehow) controls the timing of DNA replication. They seem to understand this: "These observations suggest that global R-loop distribution reflects activities that have resulted in a chromosome size-dependent patterning of many aspects of genome content and activity in L. major" but elsewhere seem to emphasise the crucial role of R-loops "DNA replication timing is intimately linked with Leishmania genome plasticity through R-loops". I invite the authors to comment on whether they think R-loops are an especially significant part of genome patterning. Are they a cause of anything, or simply an effect of the 'activities' that produce genome patterning? I invite them to point to relevant statement in the manuscript.

Though the reviewer is correct that we provide a lot of correlations (Fig.2), we feel this is valuable, given the range of features that reflect chromosome size-related replication timing in this unusual eukaryote. The

new data we have provided showing that mutations emerge after loss of RNase H1 at locations where RNase H1 normally binds R-loops (Fig.3, Fig.5, new Fig.6), suggests to us that chromatin state and DNA replication-related activities at these sites can drive genome variability. Although we cannot say for certain if R-loops are the cause or an effect, we present potential scenarios in the discussion:

'One explanation that might be considered as the basis for the connection between these aspects of the genome is reduced nucleosome density in the larger chromosomes, as this may allow greater levels of R-loops to accumulate, meaning the global correlation between chromatin status and R-loops reflects the localised coordination of these features at inter-CDS regions. Such decreased chromatin compaction may also allow for better resolution of G4s structures in the larger chromosomes, leading to reduction in the replication initiation activity detected by SNS-seq (which is associated with the presence of G4s)¹⁴⁰. However, what aspect of Leishmania biology might necessitate a gradient of nucleosome density across its chromosomes is unclear.'

'...the greater abundance of R-loops as L. major chromosome size increases may reflect less efficient DNA replication as initiation events detected by SNS-seq become sparser.'

Minor points:

- The introduction needs to explain what an R-loop is and how it relates to DNA/RNA hybrid structures?

Why should a cell need to remove them?

We have added the following text:

'We show that the distribution of R-loops, a class of RNA-DNA hybrids that form on double-stranded DNA and extrude a single DNA strand, has a striking correlation with L. major chromosome length, and that loss of the enzyme RNase H1 alters both DNA replication timing and genome stability'.

- The introduction might also briefly explain what the bigger picture is here. The manuscript drives at the idea that the timing of replication within genomes is flexible and adaptive. Better say that explicitly.

For Leishmania, and indeed any protist, we do not know if DNA replication timing is flexible and adaptive; this is the first study that attempts to examine the basis of timing in any protist, and so we would prefer not to make such predictions.

- l106, typo, schizogony

Corrected, thank you.

- l363: typo, aneuploid

Corrected, thank you

- I1394-5: “We have previously observed that SNP occurrence among chromosomes correlates with their length” – what is meant by ‘occurrence’ and what form of correlation is observed?

Clarified, thank you:

‘We have previously observed that the density of new SNPs that arise during L. major growth correlates with chromosome length¹³⁵,...’

- Throughout, be more precise in describing what is being measured. DRIP-seq ‘pattern’ (I206), R-loop ‘accumulation’ (I215), DRIPseq ‘levels’ (I226), ‘accumulation’ (I268). Is it most precise to say ‘R-loop density’? Use a consistent term that makes it clear that the number of R-loops is corrected for chromosome size.

As we evaluate the data in a number of different ways, we do not feel there is a single term that is applicable in all cases.

Reviewer #3 (Remarks to the Author):

In this manuscript, Damasceno et al., profile both R-loop and RNase H1 genomic distributions in the Leishmania major genome and show that these accumulate in a chromosome length-dependent manner that itself is negatively correlated with replication timing. Using conditional gene KO, they further show that RNase H1 loss leads to transient growth perturbation and a flattening of the replication timing differences between short and long chromosomes. In addition, RNase H1 KO leads to increased frequencies of CNVs, Indels and SNPs.

Despite reporting some interesting results on an interesting topic, the manuscript suffers from a number of significant flaws, including a tendency to over-interpret findings. The overall message that is encapsulated in the title, namely that “R-loops acted on by RNase H1 are a determinant of chromosome length-associated DNA replication timing and genome stability in Leishmania” is unsupported by evidence. The claim R-loops are acted upon by RNase H1 is unsupported because authors do not present any overlap analysis between their R-loop maps and RNase H1 maps. We are told that they RNase H1 enriches at regions where R-loops are found to accumulate without any direct comparisons being performed. We are told that “R-loops acted upon by RNase H1 have widespread effects on L. major genome instability” without any direct high-res comparison of R-loops maps with CNV, Indels or SNPs maps. Finally, we are told that “R-loops are determinant of the replication timing” without being provided evidence of overlap between R-loops, RNase H1 and replication origins. This is all the more puzzling that the data is at hand. Overall, it is clear that RNase

H1 has a strong effect on the replication program and on genome instability. I am much less convinced that R-loops have anything to do with this. If the authors want to push the message that R-loops are involved then they need to show that 1) RNase H1 binding sites overlap strongly with (at least a subset) of R-loops; and 2) these RNase H1-bound R-loops are exactly where CNVs and InDels/SNP accumulate.

The reviewer is correct, and we recognise that these correlations are obscured in the way that we have shown the data (ie dispersed over a number of figures). As further detailed below, we have:

1. Added *new data to Fig.3*, demonstrating that RNase H1 ChIP-seq does indeed overlap with a substantial fraction of R-loops predicted by DRIP-seq;

2. Added *new data to Fig.5 and a new Fig.6*, which show that patterns of CNV and mutation (SNPs and Indels) are determined by whether or not R-loops are bound by RNase H1.

Thus, these new data show that the changes we document are due to RNase H1 action on R-loops. The data also reveal differences between cells that have recently lost RNase H1 (by DiCre excision) and full gene knockouts, suggesting there has been adaptation, as we document in the DiCre growth curves.

Major comments:

1- The authors map R-loops in *Leishmania* using DRIP-seq. A key finding is that “R-loops display pronounced accumulation at intergenic regions between coding sequences (CDSs) in polycistronic transcription units”. R-loops in yeasts and mammalian cells clearly occur co-transcriptionally. Could the authors please provide transcription data (ideally nascent) to indicate whether intergenic regions are transcribed? If not, how do they envision R-loops forming?? The authors should add correlation with transcription on their Figure 2 set of panels.

Having re-read our text, we see the wording is unclear; please accept our apologies. ‘Intergenic’ is an inaccurate term in the context of kinetoplastid gene expression. In these organisms, virtually all genes are encoded from multigene/polycistronic transcription units; ie multiple genes share a single promoter, and mRNA for individual CDSs (virtually none of which have introns) is generated by coupled trans-splicing and polyadenylation at inter-CDS regions, which is where R-loops accumulate in *L. major* (and also in *T. brucei*: Briggs et al. (2018). *Nucleic Acids Res* 46(22): 11789-11805). Thus, what we called ‘intergenic’ should be more accurately stated as inter-CDS (*we have altered the text throughout*). In addition, *we show that all these inter-CDS regions are actively transcribed by comparing nascent transcript (PRO-seq) and mature transcript (RNA-seq) signals in a polycistron of L. major chromosome 1 (new Fig.S2A) and as a metaplots around all CDSs (new Fig.S2B)*. Text changes:

‘To understand the localisation of RNA-DNA hybrids, we compared the distribution of DRIP-seq signal to a number of genome features, revealing that R-loops display pronounced accumulation at regions between coding sequences (CDSs) in polycistronic transcription units, where nascent transcription is abundant (Fig.

S2), chromatin occupancy is lower (as determined by MNase-seq)⁸³ and G-quadruplex (G4)¹¹⁰ occurrence is higher (Fig. 1B and 1C). At a finer level, R-loop accumulation in inter-CDS regions overlapped with splice leader (SL) and polyadenylation (Poly A) acceptor sites (Fig. 1B and 1C).

2- Authors initially indicate that R-loops correlate with higher G4 occurrence but motif analysis (1E) doesn't reveal the presence of that motif. Why? In fact, motif analysis seems to indicate that R-loop-associated sequence features are opposite that of G4s. Figures 2A, 2E and 2R show an inverse correlation between G4s and R-loops. This needs to be clarified. The statement that "Thus, the presence of G4s is a good predictor of R-loop accumulation at the local level, but not at a genome-wide level" (line 215) is puzzling especially given the preceding statement that "Re-analysis of G4-seq data revealed a similar and significant anticorrelation with DRIP-seq".

Thank you for this observation, which we had overlooked. We re-ran the MEME analysis using 50 nt as maximum motif width, instead of the 15 nt maximum motif width used previously. In these conditions, we still recover the previously identified motifs (Fig. 1D), but now a motif emerged that contains 4 consecutive guanidine tracks, similar to G4-forming sequences. These data are shown in a revised version of Fig. S2D, and we have added the following text:

'...we also found sequences associated with G4 formation to be enriched at DRIP-seq peaks (Fig. S2D), consistent with R-loop and G4 co-localisation.'

The identification of TCn and CCTn as R-loop-enriched motifs is very puzzling as these sequences are among the absolute worst in terms of RNA:DNA hybrid stability. R-loops in yeasts and mammalian cells have long been associated with purine-rich regions, mostly G-rich. How do the authors explain this? Have they provided evidence that a pyrimidine rich regions, when cloned in an in vitro transcription plasmid vector, can form R-loops? I doubt it. This raises significant questions. Could the issue here be one of strandedness? The maps produced here are not stranded (which is too bad). Is it possible that transcription runs antisense to that of the transcription units? This would restore purine-rich motifs as R-loop-associated instead of pyrimidine-rich motifs.

PRO-seq data from *L. major* generated by Grunebast et al (BioRxiv 10.1101/2023.11.23.568479) did not reveal any significant levels of anti-sense transcription from transcription units. However, we note that Leishmania multigene transcription units themselves can 'sense' or 'anti-sense'-oriented, and thus R-loops contained in the 'anti-sense' units are likely formed by reverse complement RNA sequences relative to the identified motif DNA sequence. To clarify this, we included the reverse complement sequences for all identified motifs (Fig. 1D and Fig. S2D). We also quantified the proportion of each of these peaks falling

within 'sense' or 'anti-sense'-oriented transcription units, and show the levels of nascent transcripts overlapping each group (Fig. S2C). Text addition:

'To examine this association further, we identified 12,219 DRIP-seq peaks across the L. major genome and used MEME¹¹² to identify DNA sequences that were enriched, which revealed two motifs composed of polypyrimidines (TC_n and CCT_n) and two other motifs composed of either TA or TG repeats (Fig. 1D). RNA-DNA hybrid formation is favoured at purine-rich sequences in other eukaryotes¹¹³. A significant proportion of the R-loop-associated TC_n and CCT_n motifs in L. major are on the antisense DNA strand within a transcription unit (Fig. S2C), suggesting that R-loops that form in these regions are composed of RNA molecules corresponding to the reverse complement genome sequence, and are also therefore found at purine-rich sequence. R-loops in T. brucei are also enriched in polypyrimidine-containing regions⁹⁷, suggesting the existence of trypanosomatid-specific mechanisms for R-loop stabilisation or resolution when they form within transcription units in the sense DNA strand.'

Finally, the observation of an anti-correlation between DRIP signals and GC fraction is very counter-intuitive. I wonder if the observation of a positive correlation with SIDER1 retroposons provides an answer to some of the questions raised here. Namely, could it be possible that DRIP detects RNA:DNA hybrids that are the products of a retrotransposition event? Previous mapping of R-loops in yeast showed that Ty element activity generates signals that can be captured in DRIP (PMID: 25357144). This may also explain the association between DRIP signals and intergenic regions. The authors should provide additional information about this possibility.

These data are now provided in new Fig.S8, where we show that there is no clear enrichment of DRIP-seq signal overlapping SIDER sequences. As for GC levels, the SIDER 1 and R-loop distributions indicate an unusual, chromosome-size dependent skew in many aspects of L. major genome composition, which are not necessarily directly functionally responsible for each other, but nonetheless deviate from what is known from model eukaryotes, including yeast.

Text addition:

'Because retrotransposition can give rise to R-loops in yeast¹¹⁶, we plotted DRIP-seq signal around SIDER1, SIDER2 and non-SIDER repeats. Despite SIDER1 repeats being more abundant in chromosomes with higher R-loops density (Fig. 2S), no localized enrichment of DRIP-seq signal overlapping these regions was detected. This suggests SIDER1 are not significant sources of R-loop generation in the larger chromosomes in this parasite (Fig. S8).'

3- One of the key claims of the MS, namely that "R-loops are acted on by RNase H1" clearly predicts that RNase H1 bindings sites should overlap with R-loops. Nowhere is a detailed overlap analysis of these two

datasets provided. This is unacceptable. DRIP-seq tracks should be shown for Figure 3B and 3C. Nowhere is it stated that R-loops correlate with AcH3 or base J, which seem to correlate very strongly with RNase H1 binding sites. The statement that “the global distribution of *L. major* RNase H1 mirrors that of the R-loops that it acts to dissolve” might be true at a bird’s eye view but the devil is in the detail and high resolution comparisons should be provided to back this key claim.

The reviewer’s criticisms are very fair, and so we have added *new comparisons of RNase H1 ChIP-seq and DRIP-seq to Fig.3: a new panel of whole-chromosome signals in 3D; metaplots showing overlap between DRIP-seq and RNase H1 ChIP-seq at SSRs (end of the polycistronic transcription units) in new panel in 3E; metaplots showing overlap between DRIP-seq and RNase H1 ChIP-seq at inter-CDS regions within PTUs in new panel 3F; metaplots detailing the extent of overlap between RNase H1 ChIP-seq signal and R-loops peaks in new panels 3G,H; and linear regression showing correlation between R-loops and RNaseH1 average levels for each chromosome in new panel 3I.*

We hope that these data now provide the high-resolution comparisons that were lacking, and provide these key observations:

1. As we discussed previously, RNase H1 acts on R-loops at sites of RNA pol II transcription initiation, which may be equivalent to class I R-loops at sites of transcription pausing, as seen in other eukaryotes.
2. Though RNase H1 accumulates at termination sites, we see no similarly localised R-loop accumulation at these loci, an observation that is consistent with what was seen for R-loops in *T. brucei* (Briggs et al. (2018). *Nucleic Acids Res* 46(22): 11789-11805) and raises unanswered functional questions.
3. RNase H1 colocalises with a subset of inter-CDS R-loops, a limitation that is related to chromatin accessibility.

Thus, the distribution of R-loops and RNase H1 in *L. major* displays both local and global correlations. We have reflected these new observations in changed text in the results and discussion:

‘Visual inspection indicated that that RNase H1-HA was most strongly enriched at the boundaries of the polycistronic transcription units (PTUs), termed strand switch regions (SSRs), co-localizing with acetylated histone H3 (AcH3)¹²¹ and β -D-glucosyl-hydroxymethyluracil (Base J)¹²², markers of transcription initiation and termination sites, respectively (Fig. 3D and 3E). Furthermore, visual comparison between RNase H1-HA ChIP-seq and DRIP-seq signals indicated considerable overlap between RNase H1-HA and R-loop peaks (Fig.3D), an observation supported by metaplots that showed elevated DRIP-seq signal around SSRs, where RNase H1-HA accumulated (Fig. 3E), and overlap of RNase H1-HA ChIP-seq and DRIP-seq signal at inter-CDS regions within PTUs (Fig. 3F). Importantly, however, such co-localisation was not uniform, since examination of RNase H1-HA ChIP-seq signal around DRIP-seq peaks suggests that ~33% of R-loop regions are RNase H1-bound, while the remaining ~67% are RNase H1-free (Fig.3G). The explanation for the difference between these two

classes of R-loops, which present similar average levels of DRIP-seq and SNS-seq signals, appears to lie in chromatin accessibility, since RNase H1-bound R-loops are located at genomic regions with lower average MNase-seq signal (Fig.3H).'

'Here, we describe mapping of RNase H1 by CHIP-seq and show that its enrichment within the PTUs correlates with DRIP-seq at many inter-CDS loci, indicating the enzyme can act on R-loops that arise during transcription elongation, albeit in a role that appears to be influenced by chromatin levels. This correlation may suggest these R-loops are the equivalent of class II RNA-DNA hybrids that form during transcription and have been described in other eukaryotes¹³⁶, though as noted above, they may be unique to kinetoplastids due to the ubiquitous need for pre-mRNA processing by trans-splicing and polyadenylation.'

*'Localisation of RNase H1 at polycistronic transcription termination sites is less easy to explain, since DRIP-seq does not suggest these SSRs are pronounced sites of similarly localised R-loop enrichment in *L. major* (this work) or *T. brucei*⁹⁷.'*

4- The observation that R-loop signals measured by S9.6 immunofluorescence increase upon HU treatment but not under any other tested conditions is interesting but raises quite a few other questions. I question the reliability of the assay. The authors purport to show that S9.6 IF works in Leishmania when it is known not to work in mammalian cells (Figure 1A). I wish I could believe this but for this, I would need to see additional controls such as RNase T1 insensitivity. I also believe the authors should add MgCl₂ to the mock treated sample in case Mg ions facilitates RNA degradation by endogenous ribonucleases; having one sample receiving 5 mM MgCl₂ and RNase H1 and the other sample receiving nothing is not sufficient. Given the concerns about S9.6 imaging, I believe the authors should perform DRIP-seq in H1 KO cells to substantiate their claims.

The suggested controls are now provided in *new Figure S1*. Addition of MgCl₂ does not alter signal detected by S9.6, which (as we note) is near-exclusively nuclear. In addition, while treatment with RNase T1 somewhat reduces the nuclear signal, the effect is substantially less than is seen with *E. coli* RNase HI treatment. It might also be noted that IF with S9.6 also gives a near-exclusively nuclear signal in *Trypanosoma brucei* (Girasol et al. (2023). Nucleic Acids Res 51(20): 11123-11141; Girasol et al. (2023). Proc Natl Acad Sci U S A 120(48): e2309306120), so there is a notable difference in avidity in two trypanosomatids relative to mammalian cells. We included the following text to describe these new experiments:

'To rule out experimental artifacts due to MgCl₂-mediated RNA degradation by endogenous nucleases or unspecific RNA detection by S9.6, we also performed immunofluorescence using the S9.6 antibody after incubating cells in the presence of MgCl₂ alone or after RNase T1 treatment (Fig. S1). No significant signal reduction was observed upon MgCl₂ incubation, and while RNase T1 treatment caused a moderate signal

reduction, this was significantly less than the signal loss seen after E. coli RNase H1 treatment (Fig.S1). Altogether, these data demonstrate the specificity of the S9.6 antibody for RNA-DNA hybrids in L. major.'

5- The observation of a flattened replication program with increased CNVs and InDels in the absence of RNase H1 is interesting. Mechanistically, however, how this arises remains obscure. If the authors want to convince that R-loops "acted upon by RNase H1" have a hand in causing genome instability, they again need to compare the CNV maps to the R-loop DRIP maps directly and at high resolution. Asking readers to visually compare Figure 5B (CNVs) to Figure 2A is totally inappropriate. It seems to me that RNase H1 has a clear hand in regulating the replication program in Leishmania (how?) but I am quite unclear that R-loops have anything to do with it. For this correlation to be better documented, authors need to show that 1) RNase H1 binding sites overlap strongly with (at least a subset) of R-loops and 2) these RNase H1-bound R-loops are exactly where CNVs and InDels/SNP accumulate. The authors have the data at hand. That they do not provide a direct comparison of these datasets when it is so relevant to their main claims is raising questions.

We (again) agree that this is a fair criticism and have addressed it. First, and as described above (comment 3), we show that there is significant local and global correlation between RNase H1-HA ChIP-seq and DRIP-seq peaks. Second, we have added new analysis to Fig.5 and new Fig.6, where we compare DRIP-seq, RNase H1 ChIP-seq and locations and levels of CNV and SNP/Indel accumulation. These data now clearly show the locations of change (CNV and mutation) are most pronounced where R-loops are bound by RNase H1. As a result, this suggestion has not only clarified the paper but added mechanistic insight, and we are grateful to the reviewer for these suggestions.

We have amended the results to discuss the new data in Figs 5 and 6:

'To test if CNV occurrence in these regions correlates with RNase H1 action on R-loops, we next compared the CNV and DRIP-seq data with RNase H1-HA ChIP-seq. Although rRNA- and SL-RNA loci showed pronounced accumulation of both RNaseH1 and R-loops, no such pronounced R-loop enrichment was observed at the subtelomeres (Fig. 5E). CNV also clearly correlated with wider regions of DRIP-seq signal enrichment in both uninduced and induced cells, and the extent of this variation was greater at those loci that were bound by RNase H1 (Fig. 5D). In KO cells, however, more pronounced CNV was seen around RNase H1-free regions than at RNase H1-bound regions. These data indicate that R-loops acted upon by RNase H1 are pronounced regions of CNV both globally and locally, though loss of the endonuclease may result in adaptation that is reflected in a changed pattern of such variation.'

'To understand how the SNPs and Indels arise, we compared SNP and InDel levels with locations of DRIP-seq peaks (Fig. 6B and 6E), which revealed that while a significant fraction of R-loops are pronounced locations of these mutations, the majority are not. To understand this dichotomy, we next compared DRIP-seq and

RNase H1-HA CHIP-seq signals at regions with and without SNP and InDel accumulation (Fig. 6C and F, respectively). This analysis revealed that R-loops that resulted in increased mutation levels showed greater recruitment of RNase H1 than those without mutation, a difference that was associated with increased chromatin accessibility (MNase-seq) and predicted DNA replication initiation (SNS-seq).'

Other comments:

- The manuscript is very richly referenced (169 in total!). The Introduction is very detailed – is this all necessary? The authors map R-loops in Leishmania using DRIP-seq. Ironically, a reference to the original DRIP-seq paper is not provided.

Reference added

- In reference 87, this work only shows that R-loops occur at sites of rearrangements in CSR, not that they are required. The role, if any, of R-loops in CSR remains controversial. Please adjust.

Changed: '*...R-loops are linked to targeted...*'

- Figure 1D depicts the R-loop peak length frequency distribution. This most likely reflects the length distribution of the DNA fragments generated by restriction enzymes, not the length of R-loops. In DRIP-based assays, it is well known that the entire fragment is IP'ed and that the resolution of the method is dictated by the enzymes used for genome fragmentation. I suggest removing, or properly explaining.

We have removed the panel.

- On figure 3A, the authors need to quantify the overlap between RNase H1 and EdU signals for at least 50 independent cells to conclude that RNase H1 “showed a higher abundance in non-replicating cells”. Showing two cells is not sufficient or rigorous.

This has been added as a new panel (C) to Fig.3

- What is being shown at the bottom of Figure 4H? I thought Leishmania chromosome were replicated from a single-origin per chromosome located to subtelomeres (Introduction). Are the PTUs shown below subtelomeric? If not, how were they chosen? If origins map to SSRs, which are common, then there isn't a single origin per chromosome. Please indicate a scale! Are we looking at a whole chromosome? This is confusing. How are early and late-replicating regions defined?

This is a whole chromosome and, as we have summarised in the introduction, the main MFA-seq peak in each chromosome always localises to one (of several) SSRs (each of which is found between PTUs). In some

chromosomes, such as the 22, the main MFA-seq peak overlaps a SSR near the chromosome ends. A pronounced/main/highest MFA-seq peak indicates the location of the most early activated single (or cluster of) origin, but it more precisely reflects the timing of DNA replication. In the -RAP condition, it is possible to see the main MFA-seq peak overlapping the expected SSR for chromosome 22. The extra peaks detected in +RAP conditions and in KO cells indicate other regions are being replicated earlier when compared to -RAP.

We have adjusted the legend to explain this. New Figure S13 explains how early and late-replicating features are defined.

Response to reviewer

Reviewer #3 (Remarks to the Author):

I appreciate the authors' answers to my queries and believe the manuscript has been significantly improved as a result.

I am still puzzled by the sequence dependence of R-loops in inter-CDS regions. While the new data and clarified language helps, the legend to Figure S2C should be significantly improved prior to acceptance. As it is, I do not know what I am looking at (what are the %ages referring to? Why the two distinct arrows to the right and left? How do the motifs relate to the inter-CSD regions? Shouldn't the motifs appear in a reverse complement manner and order for the right and left arrows?). This is important since R-loop formation over pyrimidine-rich sequences would really be a departure for the R-loop field in terms of nucleic acid properties. If true, this may point to specific mechanisms at play to either form or stabilize these unusual R-loops.

The mutual relationships, at least in terms of localization, between RNase H1 and R-loops have been clarified. Significant questions surrounding the striking enrichment of RNase H1 at SSRs together with base J and Ach3 remain. Similarly, the observation that a majority of DRIP peaks are RNase H1 free raise questions. I suppose these could be explored in the future and feel confident that the present study now presents a more complete set of observations.

I particularly appreciate the improvements to the section on RNase H1 binding, R-loops and CNVs, InDels, SNPs. The manuscript is stronger as a result.

We thank the reviewer for their comments, and for their valuable time in evaluating our manuscript. We have amended the legend to Supplementary Figure 2 to clarify the details mentioned.